# RETHINKING INVARIANCE IN IN-CONTEXT LEARNING

**Lizhe Fang**[1]* **Yifei Wang**[2]* **Khashayar Gatmiry**[2] **Lei Fang**[3] **Yisen Wang**[1,4]†

[1] State Key Lab of General Artificial Intelligence,
   School of Intelligence Science and Technology, Peking University
[2] MIT CSAIL
[3] School of Economics, Peking University
[4] Institute for Artificial Intelligence, Peking University

## ABSTRACT

In-Context Learning (ICL) has emerged as a pivotal capability of auto-regressive large language models, yet it is hindered by a notable sensitivity to the ordering of context examples regardless of their mutual independence. To address this issue, recent studies have introduced several variant algorithms of ICL that achieve permutation invariance. However, many of these do not exhibit comparable performance with the standard auto-regressive ICL algorithm. In this work, we identify two crucial elements in the design of an invariant ICL algorithm: information non-leakage and context interdependence, which are not simultaneously achieved by any of the existing methods. These investigations lead us to the proposed *Invariant ICL (InvICL)*, a methodology designed to achieve invariance in ICL while ensuring the two properties. Empirically, our findings reveal that InvICL surpasses previous models, both invariant and non-invariant, in most benchmark datasets, showcasing superior generalization capabilities across varying input lengths. Code is available at https://github.com/PKU-ML/InvICL.

## 1 INTRODUCTION

In-Context Learning (ICL) has shown to be a key emergent property of large language models (LLMs) (Brown et al., 2020). By utilizing a sequence of examples as the context, LLMs can be adapted quickly and accurately to new tasks without parameter tuning (Wang et al., 2024; Kossen et al., 2024; Wang et al., 2025). Despite its impressive potential, ICL exhibits a crucially unusual behavior: *sensitivity to the order of context examples* (Lu et al., 2022; Zhao et al., 2021; Xie et al., 2021; Agrawal et al., 2022). Although context examples are independent, the order in which they are presented can dramatically influence ICL predictions, with variations from about 90% to 50% on the SST-2 dataset (Lu et al., 2022).

It is easy to note that the *auto-regressive* (AR) nature of LLMs is the root of order sensitivity. AR-LLMs often utilize a so-called *causal mask* in the attention module, which breaks the permutation invariance property of the de facto Transformer architecture[1]. As the context examples are intrinsically equivalent under different permutations, a model that respects this data symmetry tends to enhance both learning and generalization (Sokolić et al., 2016; Bietti et al., 2021; Tahmasebi & Jegelka, 2023). Therefore, recent works have proposed several variant algorithms of ICL to achieve the invariance by modifying the Transformer architecture (*e.g.,* Prefix ICL (Raffel et al., 2020), PCW (Ratner et al., 2022), and BatchICL (Zhang et al., 2024)). However, they often perform *inferior* to non-invariant counterparts like AR ICL, as we extensively observed in practice shown in Figure 1.

We note that although desirable, the invariance property alone is insufficient for good ICL performance (*e.g.,* a model with constant output $f(\cdot) = c$ is invariant yet provides no useful information). Therefore, to ensure the performance of ICL, we need to satisfy the following two properties while making ICL invariant: **1) Information Non-leakage**: it prevents the query from accessing its answer, thereby avoiding shortcuts and enabling dense learning signals for ICL by allowing the prediction of every context example in the input. **2) Context Interdependence**: Each context example interacts with all preceding examples. As the sequence lengthens, more information is provided, thereby enhancing

---

*Equal Contribution.
†Corresponding Author: Yisen Wang (yisen.wang@pku.edu.cn).
[1]Besides, sequential positional encodings (PEs) of the prompt also introduce order sensitivity.

prediction accuracy. However, existing methods more or less compromise these properties when making ICL invariant (Table 1), resulting in the lack of a well-performing invariant ICL method.

Motivated by the analysis above, we design an effective **Invariant In-context Learning (InvICL)** algorithm that maintains these essential properties, ensuring both invariance and high performance. InvICL addresses the issue of order sensitivity (invariance), not only avoiding information leakage but also enhancing context interdependence beyond what is achievable with AR-LLMs. To facilitate practical implementation, we also develop a fully parallel version of InvICL, capable of obtaining all Leave-One-Out (LOO) embeddings and predictions in a single forward pass using a novel LOO-type attention mask. Empirically, InvICL outperforms existing invariant ICL versions, and even surpasses AR-ICL (non-invariant) on most tasks of both synthetic and real-world datasets.

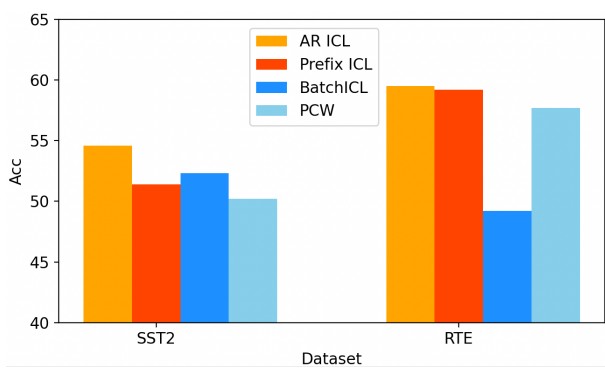

Figure 1: Performance of existing ICL algorithms under the settings of (Zhang et al., 2024), including autoregressive (AR) ICL, Prefix ICL (Raffel et al., 2020), BatchICL (Zhang et al., 2024) and PCW (Ratner et al., 2022). Task prompts are removed for fair comparison.

We summarize our contributions as follows:

- We undertake a comprehensive exploration into designing invariant ICL algorithms, highlighting the importance of preserving information non-leakage and context interdependence.
- We propose InvICL, which synergizes the goals of invariant ICL algorithms by utilizing leave-one-out embeddings to achieve invariant predictions and information non-leakage while maximizing context interdependence.
- Empirically, InvICL indeed achieves superior performance across a range of tasks on both synthetic and real-world datasets.

Table 1: Comparisons of different ICL types (details in Section 2) on permutation invariance, information non-leakage, context interdependence, and performance.

| ICL Type | Invariance | Non-leakage | Interdependence | Performance |
|---|---|---|---|---|
| Auto-regressive | ✗ | ✓ | ✓(partial) | A (baseline) |
| Prefix (full attn.) | ✓ | ✗ | ✓ | A- |
| Bag-of-Examples | ✓ | ✓ | ✗ | A- |
| **InvICL (ours)** | ✓ | ✓ | ✓ | A/A+ |

## 2 PRELIMINARIES

Consider a classification task with a few *i.i.d.* training examples $D = \{\tilde{\mathbf{x}}_i := (\mathbf{x}_i, \mathbf{y}_i)\}_{i=1}^n$, where $\mathbf{x}_i$ denotes the input and $\mathbf{y}_i$ denotes the classification target. An ICL algorithm $f$ takes these training examples (*a.k.a.* context examples) together as input and then predicts a new test example $\mathbf{x}_t$. A general formulation of $f$ is

$$[\hat{\mathbf{y}}_1, \ldots, \hat{\mathbf{y}}_n, \hat{\mathbf{y}}_t] = f(\mathbf{x}_i, \mathbf{y}_i, \ldots, \mathbf{x}_n, \mathbf{y}_n, \mathbf{x}_t), \tag{1}$$

where $\hat{\mathbf{y}}_i$ denotes the label prediction for $\mathbf{x}_i$. Note the predictions for context example, $\{\hat{\mathbf{y}}_i\}_{i=1}^n$, are optional but they are generally available for AR-LLMs.

A popular model choice for ICL is Transformer (Vaswani et al., 2017), where the self-attention layer is the elementary module. Denote $\mathbf{H} = (\mathbf{h}_1, ..., \mathbf{h}_n)^\top$ be the input hidden state of a self-attention layer, it outputs

$$\mathbf{H} \leftarrow \mathbf{H} + \mathbf{AHW}_v \mathbf{P}, \text{ where } \mathbf{A} = \text{softmax}\left(\mathbf{HW}_q(\mathbf{HW}_k)^\top + \mathbf{M}\right). \tag{2}$$

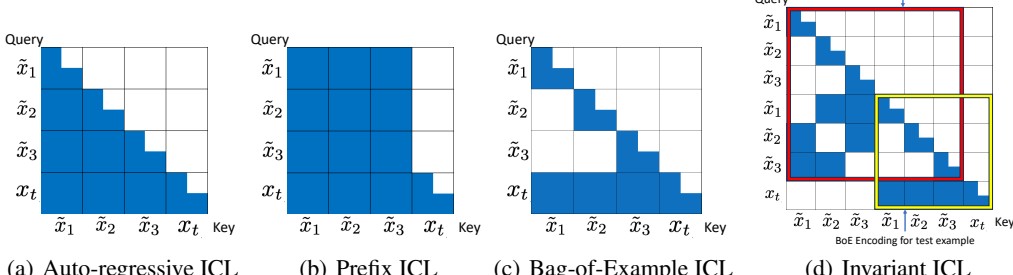

Figure 2: The attention masks of four types of ICL, corresponding to different types of ICL methods.

where $\mathbf{W}_q, \mathbf{W}_k, \mathbf{W}_v, \mathbf{P}$ denotes the query, key, value, and projector matrices, respectively. $\mathbf{M} \in \{0, -\infty\}^{n \times n}$ is an attention mask. For a standard (or full) self-attention layer, $\mathbf{M}$ is a zero matrix, while a causal self-attention layer utilizes the following causal mask:

$$\mathbf{M} = \begin{pmatrix} 0 & -\infty & \cdots & -\infty \\ 0 & 0 & \cdots & -\infty \\ \vdots & \vdots & \ddots & \vdots \\ 0 & 0 & \cdots & 0 \end{pmatrix}. \tag{3}$$

As a result, the softmax attention $\mathbf{A}$ only has nonzero weights in its lower triangular terms. Notably, the form of Eq. (2) can be generalized to other attention types, as will be discussed later.

Revisiting existing Transformer-based ICL algorithms, they can be categorized into three families depending on their aggregation scheme over the context examples: 1) Auto-regressive ICL, 2) Prefix ICL, and 3) Bag-of-Example ICL.

**Auto-regressive ICL (AR ICL).** A naive way to perform ICL is to adopt the original auto-regressive Transformer (Radford et al., 2018), which admits the following aggregation rules

$$\mathbf{h}_{\mathbf{x}_k} \leftarrow \mathrm{aggr} \left\{ \{ (\mathbf{h}_{\mathbf{x}_i}, \mathbf{h}_{\mathbf{y}_i}) \}_{i=1}^{k-1}, \mathbf{h}_{\mathbf{x}_k} \right\}, k \in [n+1], \tag{4}$$

where $\mathbf{h}_{\mathbf{x}_i}, \mathbf{h}_{\mathbf{y}_i}, \mathbf{h}_k$ denote the encodings of $\mathbf{x}_i, \mathbf{y}_i, (\mathbf{x}_k, \mathbf{y}_k)$, respectively. Here we let $\mathbf{x}_{n+1} := \mathbf{x}_t$ for notation simplicity. Therefore, every example $\mathbf{h}_k$ only attends to the previous ones $\mathbf{h}_{\leq k} = \{\mathbf{h}_1, \ldots, \mathbf{h}_k\}$, which introduces a sequential order to the input examples. As former examples have a smaller context, later examples in the sequence enjoy higher accuracy, as shown in Liu et al. (2022); Wu et al. (2022). Figure 2(a) illustrates the implementation by applying a causal mask $\mathbf{M}$, which is exactly the form in Eq. (3).

**Prefix ICL.** To fully utilize the information of every context example, the causal mask is discarded in Prefix LM (Raffel et al., 2020). Therefore, it aggregates over all context examples as

$$\mathbf{h}_{\mathbf{x}_k} \leftarrow \mathrm{aggr} \left\{ \{ (\mathbf{h}_{\mathbf{x}_i}, \mathbf{h}_{\mathbf{y}_i}) \}_{i=1}^{n} \right\}, \forall \, k \in [n]; \tag{5a}$$

$$\mathbf{h}_{\mathbf{x}_t} \leftarrow \mathrm{aggr} \left\{ \{ (\mathbf{h}_{\mathbf{x}_i}, \mathbf{h}_{\mathbf{y}_i}) \}_{i=1}^{n}, \mathbf{h}_{\mathbf{x}_t} \right\}. \tag{5b}$$

Figure 2(b) illustrates the implementation by modifying the attention mask $\mathbf{M}$ in Eq. (2), where it utilize full attention among the context examples $\{\tilde{\mathbf{x}}_i\}_{i=1}^{n}$ and causal attention on the test example $\tilde{\mathbf{x}}_t$.

**Bag-of-Example ICL (BoE ICL).** In addition to the two conventional designs above, there is a new variant of ICL. Methods like PCW (Ratner et al., 2022), SAICL (Cai et al., 2023), and BatchICL (Zhang et al., 2024) encode each context example $(\mathbf{x}_i, \mathbf{y}_i)$ independently (without considering other context examples), similar to the "bag-of-word" representation. Its aggregation rules can be formulated as

$$[\mathbf{h}_{\mathbf{x}_k}, \mathbf{h}_{\mathbf{y}_k}] \leftarrow \mathrm{aggr} \left\{ (\mathbf{h}_{\mathbf{x}_k}, \mathbf{h}_{\mathbf{y}_k}) \right\}, \forall \, k \in [n]; \tag{6a}$$

$$\mathbf{h}_{\mathbf{x}_t} \leftarrow \mathrm{aggr} \left\{ \{ \mathbf{h}_{\mathbf{x}_i}, \mathbf{h}_{\mathbf{y}_i}) \}_{i=1}^{n}, \mathbf{h}_{\mathbf{x}_t} \right\}. \tag{6b}$$

Figure 2(c) illustrates an implementation (PCW (Ratner et al., 2022)) by modifying the attention mask $\mathbf{M}$. It restricts attention to occur only within each context example $\tilde{\mathbf{x}}_i, i \in [n]$, preventing cross-attentions between them, while retaining attention between the test example $\tilde{\mathbf{x}}_t$ and context examples.

# 3 The Proposed Invariant In-context Learning (InvICL)

We begin with formalizing the desiderata of invariant ICL (Section 3.1), and explore how to meet all these desiderata (Section 3.2). Next, we introduce how to implement our proposed InvICL method in practice (Sections 3.3).

## 3.1 Invariant ICL and Its Desiderata

We begin with a formal characterization of three important desiderata in invariant in-context learning.

**1) Invariance.** In an ICL task, we have the prior knowledge of data symmetry that the $n$ context examples $\tilde{\mathbf{x}}_i$ are independently identical distributed (*i.i.d.*). We define an ICL algorithm that preserves this symmetry property as an invariant ICL algorithm:

**Definition 3.1.** An ICL algorithm $f$ is said to be *(permutation) invariant* if its last prediction $f_t$ satisfies $f_t(\tilde{\mathbf{x}}_1, ..., \tilde{\mathbf{x}}_n, \mathbf{x}_t) = f_t(\tilde{\mathbf{x}}_{i_1}, ..., \tilde{\mathbf{x}}_{i_n}, \mathbf{x}_t)$ for any $(i_1, \ldots, i_n) \in S_n$, a permutation of $[n] = \{1, 2, \ldots, n\}$.

**2) Information Non-leakage.** During training, AR-LLMs learn to dynamically predict each intermediate context example $\mathbf{x}_i$ based on its previous tokens $\mathbf{x}_{<i}$ as its own context, leading to $n$ prediction tasks that provide rich learning signals for ICL. To achieve this, an essential architectural inductive bias is the causal mask, which ensures that the prediction of each query $\mathbf{x}_i$, such as $\hat{\mathbf{y}}_i$, does not have access to its ground-truth answer $\mathbf{y}_i$; otherwise, the prediction task would become trivial. We believe this principle should be generally adhered to when designing ICL algorithms. We name this the *information non-leakage* principle, formally described below.

**Definition 3.2.** An ICL algorithm $f$ has **no information leakage** if its prediction of every example $\mathbf{x}_i$ is *invariant* to its label $\mathbf{y}_i$ (with others fixed), *i.e.,* $f(\ldots, \mathbf{x}_i, \mathbf{y}_i, \ldots)_i = f(\ldots, \mathbf{x}_i, \mathbf{y}'_i, \ldots)_i$ holds for any two labels $\mathbf{y}_i, \mathbf{y}'_i \in \mathcal{Y}$, where $f(\cdot)_i$ denotes the $i$-th element of $f(\cdot)$.

**3) Context Interdependence.** Another advantage of AR-LLMs is that they allow the encoding of each example $\mathbf{x}_i$ to depend on other (previous) examples. These examples provide the context for better encoding of $\mathbf{x}_i$, which in turn improves the prediction of future examples when $\mathbf{x}_i$ serves as their context. We name this property as context interdependence. Unlike the information non-leakage principle, this property requires that the prediction of each example $\mathbf{x}_i$ should flexibly depend on as many other context examples as possible.

**Definition 3.3.** An ICL algorithm $f$ is **context-interdependent** for $\mathbf{x}_i$ if the prediction of $\mathbf{x}_i$ is *dependent* on other examples. Formally, for any $j \neq i$, there exists $(\mathbf{x}'_j, \mathbf{y}'_j)$ such that $f(\ldots, \mathbf{x}_i, \mathbf{y}_i, \ldots, \mathbf{x}_j, \mathbf{y}_j, \ldots)_i \neq f(\ldots, \mathbf{x}_i, \mathbf{y}_i, \ldots, \mathbf{x}'_j, \mathbf{y}'_j, \ldots)_i$ with other examples fixed.

**Limitations of Previous Methods.** Through a close examination, we find that no existing ICL methods satisfy all these principles: 1) AR ICL avoids information leakage and has partial context interdependence, but its sequential structure breaks permutation invariance; 2) Prefix ICL maintains permutation invariance and full context interdependence, but it leaks information; 3) BoE ICL achieves permutation invariance and prevents information leakage through independent encoding, but it sacrifices context interdependence and limits the flexibility of context representations. These properties are summarized in Table 1. Motivated by the limitations of previous methods, we aim to design an ICL algorithm that achieves all three properties.

## 3.2 A Principled Design of Invariant ICL

In this section, we explore how to design ICL algorithms that preserve all three principles: permutation invariance, information non-leakage, and context interdependence. In a Transformer, the only interaction among different examples occurs in the self-attention layer (Eq. (2)). Conceptually, self-attention can be viewed as a message-passing scheme on a digraph of $n$ examples, denoted as $G$, with the adjacency matrix defined by the attention score matrix $\mathbf{A} \in \mathbb{R}^{n \times n}$:

$$\mathbf{A} = \text{softmax}\left(\mathbf{H}\mathbf{W}_q(\mathbf{H}\mathbf{W}_k)^\top + \mathbf{M}\right), \tag{7}$$

where $\mathbf{M} \in \{0, -\infty\}^{n \times n}$ is a constant mask matrix. Under this definition, $\mathbf{A}_{ij}$ represents the message passed from the $j$-th example to the $i$-th example. The message passing then updates with $\mathbf{H} \leftarrow \mathbf{A}\mathbf{H}$ (informal) using $\mathbf{A}$ as the propagation matrix. Here, we only consider the graph of context examples $\{\tilde{\mathbf{x}}_i\}_{i=1}^n$, as we always want the test example $\tilde{\mathbf{x}}_t$ to be fully aware of the context examples,

making these edges trivial. A straightforward way to design invariant ICL algorithm, commonly used by existing works (such as Prefix ICL (Raffel et al., 2020), PCW (Ratner et al., 2022), and SAICL (Cai et al., 2023)), is to modify the attention mask $\mathbf{M}$ as it is the only controllable factor in Eq. (7). Therefore, in the following, we discuss how to design the attention mask $\mathbf{M}$ to meet these desiderata. All the proofs are in Appendix D.

**Permutation Invariance by Three-choice Mask.** Intuitively, permutation invariance requires the attention mask $\mathbf{M}$ to exhibit some form of symmetry. Notably, both the Prefix and BoE masks (Figure 2(b, c)) satisfy permutation invariance, while the causal mask (Figure 2(a)) does not. The following proposition explores whether other attention masks can also achieve this property.

**Proposition 3.4.** *Given an input matrix $\mathbf{H} = (\mathbf{h}_1, ..., \mathbf{h}_n)^\top \in \mathbb{R}^{n \times d}$ with the features of the context examples only. The permutation invariance of ICL outputs (Definition 3.1) holds **if and only if** the attention mask on the context examples, $\mathbf{M}$, belongs to $\mathcal{M} = \{\mathbf{M}_1, \mathbf{M}_2, \mathbf{0}\}$, where*

$$\mathbf{M}_1 = \begin{pmatrix} 0 & -\infty & \cdots & -\infty \\ -\infty & 0 & \cdots & -\infty \\ \vdots & \vdots & \ddots & \vdots \\ -\infty & -\infty & \cdots & 0 \end{pmatrix}, \mathbf{M}_2 = \begin{pmatrix} -\infty & 0 & \cdots & 0 \\ 0 & -\infty & \cdots & 0 \\ \vdots & \vdots & \ddots & \vdots \\ 0 & 0 & \cdots & -\infty \end{pmatrix}.$$

Proposition 3.4 demonstrates that to achieve permutation invariance, the attention mask on the context example *must* fall into one of the three choices in $\mathcal{M}$: $\mathbf{0}$ corresponds to full attention in Prefix ICL; $\mathbf{M}_1$ corresponds to BoE ICL; and the attention score before softmax under $\mathbf{M}_2$ is the linear combination of that of $\mathbf{M}_1$ and $\mathbf{0}$ (only cross-attention between tokens without self-attention).

**Information Non-leakage by Lower Triangular Mask.** According to Zheng et al. (2018), ensuring information non-leakage is equivalent to guaranteeing the message-passing process through the graph is acyclic (except for self-loops). This imposes the following restriction on the attention mask $\mathbf{M}$.

**Proposition 3.5.** *An ICL algorithm realizes information non-leakage **if and only if** it is possible to reorder context examples such that the attention mask on context examples $\mathbf{M}$ is lower triangular.*

Combining the conditions for attention masks outlined in Propositions 3.4 & 3.5 (belong to $\mathcal{M}$ and lower triangular), we find that the attention mask on context examples *must* be a diagonal matrix, as concluded in the following proposition.

**Proposition 3.6.** *The message-passing scheme respects permutation invariance and information non-leakage **if and only if** the attention mask on context examples $\mathbf{M}$ is diagonal.*

Therefore, we conclude that if an ICL algorithm preserves both permutation invariance and information non-leakage, its attention mask not only can be, but *has to* be in the form depicted in Figure 2(c). Specifically, it *must* take the form of a bag-of-examples (BoE) ICL, encoding each example individually before aggregation as in Eq. (6), denoted as:

$$\mathbf{h}_{\mathbf{x}_t} \leftarrow \text{BoE}\left\{\{(\mathbf{h}_{\mathbf{x}_i}, \mathbf{h}_{\mathbf{y}_i})\}_{i=1}^n, \mathbf{h}_{\mathbf{x}_t}\right\}. \tag{8}$$

However, as discussed in Section 3.1, BoE lacks context interdependence.

**Context Interdependence through Pre-encoding.** While context interdependence cannot be implemented within a single propagation step among context examples, it can still be achieved by encoding each context example with the context of other samples, a process we term *pre-encoding*. To ensure the three principles simultaneously, the pre-encoding step must also adopt the form of a BoE ICL scheme, where it aggregates independent encodings of all other samples (*i.e.,* , a leave-one-out encoding):

$$\mathbf{h}_{\mathbf{x}_k} \leftarrow \text{BoE}\left\{\{(\bar{\mathbf{h}}_{\mathbf{x}_i}, \bar{\mathbf{h}}_{\mathbf{y}_i})\}_{i \neq k}, \mathbf{h}_{\mathbf{x}_k}\right\}, k \in [n] \tag{9}$$

where $\bar{\mathbf{h}}_{\mathbf{x}_i}, \bar{\mathbf{h}}_{\mathbf{y}_i}$ are the independent encoding (similar to Eq. (6a)). Therefore, we arrive at a two-stage ICL method as follows. First, we encode each context example with a leave-one-out (LOO) BoE encoding as in Eq. (9). Then, in the second stage, we utilize these contextual encodings to predict the test examples as in Eq. (8). This approach guarantees the three desiderata of invariant ICL.

**Symmetric Positional Encoding.** As a minor point, to ensure the symmetry of the model, it is also necessary to incorporate permutation invariance into the positional encoding. We adopt an independent position encoding scheme that treats each example as an independent sequence. It is also applicable to BoE ICL and Prefix ICL for ensuring permutation invariance. Details in Appendix A.1.

Finally, we reach our proposed **InvICL (Invariant In-context Learning)**. The propagation process for InvICL is outlined in Algorithm 1, where $\mathbf{h}_{\mathbf{x}_i}^{(k)}$ is the encoding of $\mathbf{x}_i$ at the $k$-th layer of Transformer.

---

**Algorithm 1** Invariant In-context Learning

---

**Require:** $\{(\mathbf{h}_{\mathbf{x}_i}^{(0)}, \mathbf{h}_{\mathbf{y}_i}^{(0)})\}_{i=1}^n$: embedding of context examples; $\mathbf{h}_{\mathbf{x}_t}^{(0)}$: embedding of the ICL query
1: **for** $k = 1$ to *#TransformerLayers* **do**
2:     **for** $i = 1$ to $n$ **do**
3:         Compute the independent encoding of context examples: $(\bar{\mathbf{h}}_{\mathbf{x}_i}^{(k)}, \bar{\mathbf{h}}_{\mathbf{y}_i}^{(k)}) = \mathrm{aggr}\{(\bar{\mathbf{h}}_{\mathbf{x}_i}^{(k-1)}, \bar{\mathbf{h}}_{\mathbf{y}_i}^{(k-1)})\}$ (where $\bar{\mathbf{h}}_{\mathbf{x}_i}^{(0)} = \mathbf{h}_{\mathbf{x}_i}^{(0)}$)
4:     **end for**
5:     **for** $i = 1$ to $n$ **do**
6:         Compute the leave-one-out pre-encoding of the $i$-th context example: $(\mathbf{h}_{\mathbf{x}_i}^{(k)}, \mathbf{h}_{\mathbf{y}_i}^{(k)}) = \mathrm{aggr}\{\{(\bar{\mathbf{h}}_{\mathbf{x}_j}^{(k-1)}, \bar{\mathbf{h}}_{\mathbf{y}_j}^{(k-1)})\}_{j \neq i}, \mathbf{h}_{\mathbf{x}_i}^{(k-1)}\}$
7:     **end for**
8:     Update $\mathbf{h}_{\mathbf{x}_t}^{(k)} = \mathrm{aggr}\{\{(\mathbf{h}_{\mathbf{x}_i}^{(k-1)}, \mathbf{h}_{\mathbf{y}_i}^{(k-1)})\}_{i=1}^n\}$
9: **end for**

---

### 3.3 PARALLEL IMPLEMENTATION

In Section 3.2, we have developed a truly invariant ICL algorithm achieving the three desiderata. However, a significant drawback of the encoding scheme in Algorithm 1 is its computational cost. For each sequence of $n$ context examples, it requires $n$ LOO forward passes to pre-encode each example, plus an additional forward pass for the final prediction. This results in a total of $n + 1$ forward passes for a single prediction. In contrast, AR ICL, BoE ICL, and Prefix ICL can all be implemented in parallel using a single forward pass by modifying the attention mask to the form illustrated in Figure 2(a, b, c).

**Parallel Computation via Unrolling.** To address the computational cost issue, we propose a parallel implementation for InvICL, leveraging the chain-of-thought idea from LLM reasoning (Wei et al., 2022). While implementing InvICL within a single forward pass of the input sequence $(\tilde{\mathbf{x}}_1, ..., \tilde{\mathbf{x}}_n)$ is challenging, this difficulty can be overcome by unrolling the input sequence twice. As illustrated in Figure 2(d), we duplicate the context examples twice as $(\tilde{\mathbf{x}}_1, ..., \tilde{\mathbf{x}}_n, \tilde{\mathbf{x}}_1, ..., \tilde{\mathbf{x}}_n, \mathbf{x}_t)$ and perform a two-step forward process *in parallel* to encode the context examples. In the first step, we perform a BoE-style encoding of each context example ($\bar{\mathbf{h}}_i^{(k)}$ in Algorithm 1). In the second step, we apply a LOO-style attention mask to obtain the LOO encodings of each example ($\mathbf{h}_i^{(k)}$ in Algorithm 1) that are aware of all other context examples. At last, we use the LOO encodings $\{\mathbf{h}_i^{(k)}\}$ to predict the test example $\mathbf{x}_t$. This unrolling scheme enables us to accomplish InvICL in a single forward pass, which results in the same complexity order $O(n^2)$ as the baselines.

## 4 EXPERIMENTS

### 4.1 SYNTHETIC SCENARIO

To evaluate the in-context capability of InvICL, we conduct a series of experiments inspired by Garg et al. (2022). Taking the linear regression task for example, we train a model to perform linear regression using in-context learning, *i.e.,* the model takes the sequence $(\mathbf{x}_1, g(\mathbf{x}_1), ..., \mathbf{x}_n, g(\mathbf{x}_n), \mathbf{x}_t)$ as input and predicts $g(\mathbf{x}_t)$ where $g$ is a linear function. Detailed experimental settings are provided in Appendix A.3. We compared the ICL performance across four models: 1) Auto-regressive (AR) (Radford et al., 2019); 2) Prefix (Raffel et al., 2020); 3) Bag-of-Examples (BoE) (Ratner et al., 2022); and 4) NoPE (*i.e.,* removing the positional encoding) (Kazemnejad et al., 2024). The MSE loss was reported for models trained over various epochs, as illustrated in Figure 3. The key insights from our experiments are as follows:

- **InvICL Converges Fast.** At 50k epochs, only InvICL demonstrates good ICL performance (Figure 3(a)), while other models perform well at later epochs (Figure 3(b)).

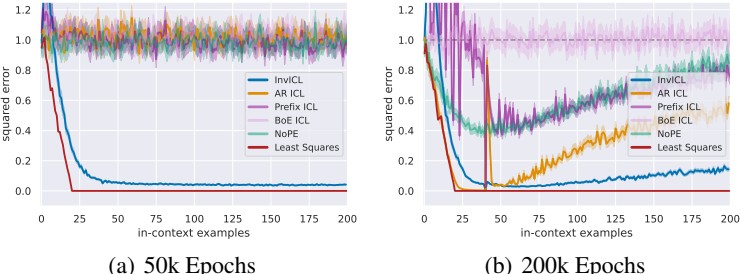

|  |  |
|:---:|:---:|
| (a) 50k Epochs | (b) 200k Epochs |

Figure 3: ICL performance of different models that are trained with **(a)** 50k epochs and **(b)** 200k epochs. "Least Squares" is the optimal algorithm for the linear regression task.

- **InvICL Has a Strong Length Extrapolation Ability.** The models are trained with a sequence length of 40. As shown in Figure 3(b), when the test sequence length exceeds 40, it is clearly that InvICL > AR ICL > Prefix ICL ≈ NoPE > BoE ICL in terms of performance. This indicates the strong length generalization capability of InvICL. On one hand, this result confirms the conventional conclusion that a model that respects the data symmetry enjoys better generalization capability. On the other hand, it highlights that preventing information leakage and maintaining context interdependence are crucial for an invariant ICL algorithm.

We further conduct experiments in out-of-distribution tasks and other function settings in Appendix B.1, and present the trend of loss as it changes with the training epochs in Appendix B.4. Both experiments demonstrate that InvICL's out-of-distribution in-context performance consistently outperforms AR ICL. Additionally, in Appendix B.5, we conduct linear probing experiments to further demonstrate how the architecture of InvICL impacts the model's internal representations.

## 4.2 REAL-WORLD DATASETS

In this part, we conduct experiments to evaluate the capacity of InvICL on real-world datasets. Since ICL tasks are generally different from the pertaining one and some ICL methods introduce new masking schemes for aggregation (significantly different from the masking in pretrained model), a short finetuning of the pretrained model on the ICL tasks using these new ICL methods is necessary to fully utilize the pretrained model's capacity for ICL (Min et al., 2022b; Wei et al., 2021; Iyer et al., 2022; Cai et al., 2023). Here, we follow MetaICL (Min et al., 2022b) to do the short finetuning and evaluation.

As in MetaICL, we utilize 142 tasks including text classification, question answering (QA), natural language inference (NLI), and paraphrase detection. For each training iteration, we first sample a task $\mathcal{T}_i$ from the $C$ meta-training tasks, and then sample $k+1$ training examples $(\mathbf{x}_1, \mathbf{y}_1), ..., (\mathbf{x}_{k+1}, \mathbf{y}_{k+1})$ from $\mathcal{T}_i$. Given the model parameter $\theta$, the training objective is maximizing prediction accuracy of $\mathbf{y}_{k+1}$ under the formatting of ICL: $\max_\theta \mathcal{L}_{\text{CE}}(\hat{\mathbf{y}}_{k+1}, \mathbf{y}_{k+1})$, where $\mathcal{L}_{\text{CE}}$ is the cross-entropy loss, and $\hat{\mathbf{y}}_{k+1}$ is the in-context prediction defined in Eq. (1). We evaluate the meta-trained models on the 7 settings of MetaICL. For each setting, we test two cases: 1) all target tasks; 2) target tasks in the training unseen domains (OOD generalization). More details are in Appendix A.4.

**Baselines.** Following MetaICL, we use GPT-2 Large (762M) (Radford et al., 2019) as base model, and also includes GPT-Neo 2.7B (Black et al., 2021) and Pythia-2.8B (Biderman et al., 2023) (Appendix B.2). For non-invariant methods, we select AR ICL (Radford et al., 2019) and NoPE[2] (Kazemnejad et al., 2024). For invariant methods, we select Prefix ICL (Raffel et al., 2020) and three types of BoE ICL (Appendix A.2): PCW (Ratner et al., 2022), SAICL (Cai et al., 2023), and BatchICL (Zhang et al., 2024). We adopt 8 context examples for training and evaluation.

**Results.** As shown in Table 2, compared to non-invariant methods, InvICL outperforms in 4 out of 7 tasks in the "*All target task*" setting and all the 7 tasks in the "*Target tasks in unseen domains*" setting. This indicates that permutation invariance is indeed an crucial property for ICL algorithm, which incorporate the inductive bias of symmetry into the model architectures, resulting in an extraordinary improvement on performance, especially when generalizing to OOD tasks.

---

[2]Although NoPE alone is invariant, it still utilizes an auto-regressive LLM which breaks the invariance.

Table 2: The in-context learning performance with language models based on GPT-2 Large. We changed the causal mask and positional encoding to implement different types of ICL models. The models are finetuned under the framework of MetaICL (Min et al., 2022b).

| METHOD | HR →LR | CLASS →CLASS | NON-CLASS →CLASS | QA →QA | NON-QA →QA | NON-NLI →NLI | NON-PARA →PARA | AVG. |
|---|---|---|---|---|---|---|---|---|
| *Non-invariant* | | | | *All target tasks* | | | | |
| AR ICL (RADFORD ET AL., 2018) | $43.4_{\pm 0.76}$ | $\mathbf{43.4}_{\pm 1.36}$ | $40.2_{\pm 1.64}$ | $44.0_{\pm 0.22}$ | $37.9_{\pm 0.42}$ | $50.3_{\pm 0.84}$ | $34.1_{\pm 1.78}$ | $41.9_{\pm 1.15}$ |
| NOPE (KAZEMNEJAD ET AL., 2024) | $41.7_{\pm 0.47}$ | $30.0_{\pm 0.82}$ | $\mathbf{40.3}_{\pm 0.99}$ | $44.5_{\pm 0.11}$ | $36.6_{\pm 0.05}$ | $26.8_{\pm 0.68}$ | $\mathbf{38.8}_{\pm 1.49}$ | $37.0_{\pm 0.81}$ |
| *Invariant* | | | | | | | | |
| PCW (BoE) (RATNER ET AL., 2022) | $39.7_{\pm 1.30}$ | $37.7_{\pm 0.51}$ | $35.2_{\pm 0.37}$ | $40.8_{\pm 0.12}$ | $37.7_{\pm 0.30}$ | $40.7_{\pm 1.32}$ | $35.1_{\pm 1.65}$ | $38.1_{\pm 0.98}$ |
| SAICL (BoE) (CAI ET AL., 2023) | $43.4_{\pm 0.45}$ | $43.2_{\pm 0.74}$ | $37.5_{\pm 0.74}$ | $45.1_{\pm 0.15}$ | $37.6_{\pm 0.15}$ | $49.8_{\pm 2.01}$ | $33.3_{\pm 1.44}$ | $41.4_{\pm 1.03}$ |
| BATCHICL (BoE) (ZHANG ET AL., 2024) | $31.7_{\pm 0.21}$ | $25.4_{\pm 0.30}$ | $27.1_{\pm 0.22}$ | $32.2_{\pm 0.12}$ | $34.4_{\pm 0.26}$ | $28.9_{\pm 0.48}$ | $35.3_{\pm 0.97}$ | $30.7_{\pm 0.45}$ |
| PREFIX ICL (RAFFEL ET AL., 2020) | $40.3_{\pm 0.89}$ | $39.6_{\pm 0.73}$ | $35.1_{\pm 0.54}$ | $43.6_{\pm 0.12}$ | $36.8_{\pm 0.33}$ | $45.4_{\pm 1.65}$ | $34.9_{\pm 2.03}$ | $39.4_{\pm 1.11}$ |
| INVICL(OURS) | $\mathbf{45.1}_{\pm 1.31}$ | $42.9_{\pm 0.86}$ | $39.4_{\pm 0.44}$ | $\mathbf{45.3}_{\pm 0.15}$ | $\mathbf{38.3}_{\pm 0.27}$ | $\mathbf{51.6}_{\pm 0.85}$ | $34.7_{\pm 1.36}$ | $\mathbf{42.4}_{\pm 0.87}$ |
| *Non-invariant* | | | | *Target tasks in unseen domains* | | | | |
| AR ICL (RADFORD ET AL., 2018) | $31.5_{\pm 2.98}$ | $35.7_{\pm 0.50}$ | $28.1_{\pm 1.65}$ | $56.5_{\pm 0.89}$ | $39.2_{\pm 1.78}$ | $80.3_{\pm 1.80}$ | $34.1_{\pm 0.00}$ | $43.6_{\pm 1.65}$ |
| NOPE (KAZEMNEJAD ET AL., 2024) | $32.9_{\pm 1.32}$ | $23.4_{\pm 0.39}$ | $26.9_{\pm 1.44}$ | $63.6_{\pm 0.78}$ | $38.2_{\pm 0.34}$ | $33.2_{\pm 0.26}$ | $32.6_{\pm 0.16}$ | $35.8_{\pm 0.83}$ |
| *Invariant* | | | | | | | | |
| PCW (BoE) (RATNER ET AL., 2022) | $35.6_{\pm 2.54}$ | $31.3_{\pm 0.29}$ | $26.9_{\pm 1.59}$ | $65.3_{\pm 1.16}$ | $33.7_{\pm 1.21}$ | $66.7_{\pm 1.60}$ | $34.4_{\pm 0.31}$ | $42.0_{\pm 1.44}$ |
| SAICL (BoE) (CAI ET AL., 2023) | $30.7_{\pm 1.67}$ | $29.7_{\pm 1.98}$ | $26.4_{\pm 1.01}$ | $56.2_{\pm 0.50}$ | $41.5_{\pm 1.60}$ | $64.3_{\pm 2.21}$ | $37.1_{\pm 1.89}$ | $40.8_{\pm 1.65}$ |
| BATCHICL (BoE) (ZHANG ET AL., 2024) | $24.2_{\pm 0.21}$ | $22.3_{\pm 0.15}$ | $23.0_{\pm 0.11}$ | $31.9_{\pm 1.20}$ | $29.4_{\pm 0.54}$ | $37.8_{\pm 0.78}$ | $36.8_{\pm 1.02}$ | $29.3_{\pm 0.70}$ |
| PREFIX ICL (RAFFEL ET AL., 2020) | $31.0_{\pm 2.43}$ | $33.0_{\pm 1.53}$ | $\mathbf{29.6}_{\pm 2.20}$ | $63.8_{\pm 0.47}$ | $36.4_{\pm 1.29}$ | $52.6_{\pm 2.54}$ | $34.0_{\pm 0.23}$ | $40.1_{\pm 1.75}$ |
| INVICL(OURS) | $\mathbf{44.4}_{\pm 2.17}$ | $\mathbf{35.8}_{\pm 2.01}$ | $29.0_{\pm 1.99}$ | $\mathbf{67.6}_{\pm 0.22}$ | $\mathbf{42.6}_{\pm 1.53}$ | $\mathbf{81.8}_{\pm 0.65}$ | $\mathbf{37.5}_{\pm 2.30}$ | $\mathbf{48.4}_{\pm 1.72}$ |

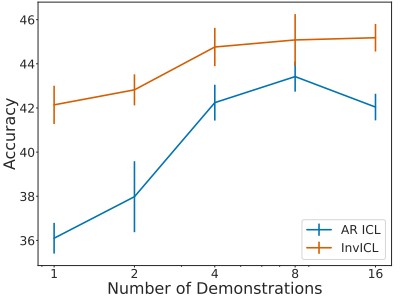

Figure 4: The length generalization behavior of InvICL and AR ICL on HR→LR setting. The models are meta-trained by sequences with 8 context examples.

Table 3: The inference time of different models.

| Method | Inference time (ms) |
|---|---|
| AR ICL | 21.9 |
| PCW (BoE ICL) | 21.7 |
| Prefix ICL | 22.0 |
| InvICL | 22.0 |

Compared to invariant methods, InvICL outperforms 5 out of 7 tasks in the "*All target task*" setting and 6 out of 7 tasks in the "*Target tasks in unseen domains*" setting. Although being permutation invariant, these baselines exhibit poor performance (none of them surpasses AR ICL on average). This highlights the crucial properties of information non-leakage and context interdependence implemented by InvICL.

**Length Generalization.** The generalization ability to different input lengths is a crucial property of the language model. In the context of ICL, the ability to adapt to varying numbers of context examples can be perceived as a dimension of its length generalization capability. However, in the main experiments, the number of context examples remains consistent throughout both the training and evaluation phases. Hence, we vary the number of context examples, as illustrated in Figure 4. We observe that InvICL is much more robust than AR ICL when the length of the test data differs from that of the training data, indicating its strong capability for length generalization.

**Computational Cost.** In Section 3.3, we claim that our parallel implementation of InvICL has the same computational complexity order as full self-attention and AR self-attention. In Table 3, we empirically verify this by evaluating the inference time of different ICL models, showing that InvICL enjoys roughly the same inference speed as other models. Besides, a question worth considering is the memory cost of InvICL since it duplicates the input sequence. We find that when the inputs size of the GPT-2 Large model increases from 512 to 1024, the GPU memory overhead increases by 14% (from 4.2 GB to 4.8GB). We consider this acceptable given the clear improvements in performance.

**Ablation Study.** In Table 4, we conduct an ablation study to demonstrate the effect of the two components of InvICL: the invariant mask and the symmetric positional encoding. The experiments show that either component is important for invariant ICL. Additionally, in Appendix B.3, we demonstrate that the effectiveness of InvICL is not due to its doubled input.

Table 4: Ablation study of invariant mask and symmetric positional encodings (PE) on ICL performance and order sensitivity.

| METHOD | HR→LR (↑) | SENSITIVITY (↓) |
|---|---|---|
| AR ICL | 43.4 | 0.25 |
| +SYM PE | $38.4_{-5.0}$ | $0.30_{+0.05}$ |
| +INV MASK | $44.8_{+1.4}$ | $0.10_{-0.15}$ |
| +BOTH (INVICL) | $45.1_{+1.7}$ | $0.00_{-0.25}$ |

**Permutation Invariance.** In Table 4, we demonstrate the permutation invariance of InvICL. Following Chen et al. (2022), we measure the order sensitivity as the frequency that the prediction is changed under random permutation. We observe that both the invariant mask and PE are important for achieving invariance, and a lower sensitivity indicates better performance.

## 5 DISCUSSION

**The Mechanism behind InvICL's Strong Length Generalization Ability.** We consider that the mechanism primarily stems from InvICL achieving invariance. As mentioned in the introduction, previous studies have found that respecting data symmetry in models helps improve generalization. For example, Sokolić et al. (2016) demonstrated that when the input data exhibits invariance under certain transformations (such as rotation or translation), utilizing an invariant classifier can achieve lower generalization error compared to a regular classifier. Bietti et al. (2021); Tahmasebi & Jegelka (2023) concluded that encoding invariances into model improves the effective number of samples, thereby enhance generalization ability. These theoretical results could help explain why InvICL demonstrates stronger length generalization ability.

**Theoretical Complexity of InvICL.** Suppose there are $n$ context examples and 1 test example (considering the examples as attention units), and let $M \in \{0, -\infty\}^{(n+1) \times (n+1)}$ be the attention mask defined in Figure 2(d). The complexity of InvICL is determined by the number of "0" elements in $M$. The attention computation for InvICL includes: 1) Independent self-encoding of the first-time input (corresponding to M[:n, :n]), which requires $n$ self-attention calculations; 2) LOO pre-encoding (corresponding to M[n: 2n, :2n]), which requires $n^2$ calculations; 3) Aggregation to the test example (corresponding to M[2n+1, n: 2n+1]), which requires $n+1$ calculations. In total, there are $n^2 + 2n + 1$ attention calculations, which is of the same order as Prefix ICL ($n^2 + 1$) and twice that of AR ICL ($n^2/2 + 3n/2 + 1$).

**The ICL Training Objective.** In the synthetic experiments, we utilize the ICL objective to train the Transformers, which does not align with how LLMs are pre-trained. However, our paper focuses on improving the ICL capability of LLMs, rather than investigating the reasons behind the emergence of ICL ability. Therefore, we train the model using the ICL objective to demonstrate that InvICL can achieve stronger ICL capability compared to traditional AR ICL. This is also aligned with the objective we use in the real-world experiments.

**Theoretical Understanding InvICL from an Optimization Perspective.** Previous studies have established the duality between ICL and the gradient descent algorithm, demonstrating that under specific parameterizations, ICL can implicitly implement gradient descent. In Appendix C, we build upon this line of research and prove that InvICL, under the same parameterizations, can also approximately perform gradient descent, thereby highlighting the theoretical potential of InvICL.

## 6 CONCLUSION

In this paper, by distilling the advantages of auto-regressive language models, we identified two additional desiderata for invariant ICL: information non-leakage and context interdependence. Since existing invariant ICL algorithms cannot achieve these desiderata simultaneously, we proposed a novel invariant ICL scheme called Invariant In-context Learning (InvICL), which accomplishes these goals concurrently. We also proposed an efficient parallel implementation of InvICL. Empirically, we show that InvICL outperforms both invariant and non-invariant ICL methods on most tasks, and demonstrates good length generalization abilities. These results sparked the unique advantages of the principled design of invariant ICL.

ACKNOWLEDGEMENT

Yisen Wang was supported by National Key R&D Program of China (2022ZD0160300), National Natural Science Foundation of China (92370129, 62376010), and Beijing Nova Program (20230484344, 20240484642). Yifei Wang was supported in part by the NSF AI Institute TILOS, and an Alexander von Humboldt Professorship.

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

# A    IMPLEMENTATION DETAILS

## A.1    SYMMETRIC POSITIONAL ENCODING

In this paper, we mainly focus on the absolute positional encoding which is used in the GPT family. As shown in Figure 5, we adopt an independent position encoding scheme that treats each example as an independent sequence, which follows the design in (Ratner et al., 2022). For each context example $\tilde{\mathbf{x}}_i$, we always allocate the positional encoding as it starts from the first position. Denote the maximal sequence length among $\tilde{\mathbf{x}}_i$ as $l_{\max}$. For the test example $\mathbf{x}_t$, we assign its positional encodings starting from the index $\ell_{\max}$. This implementation is applicable to BoE ICL, Prefix ICL, and InvICL.

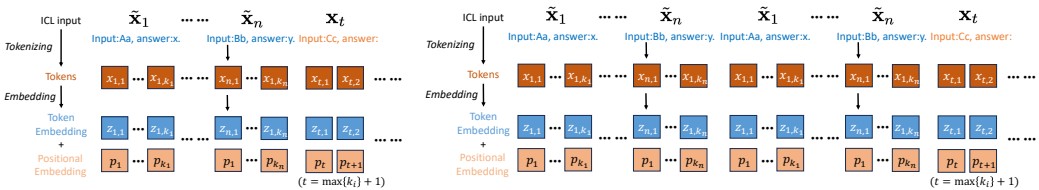

(a) Symmetric PE for standard input      (b) Symmetric PE for the duplicated input of InvICL

Figure 5: The symmetric positional encoding applied in our work. $p_i$ refers to the learned absolute positional embeddings that are added to the token embeddings at position $i$. Figure **(a)** shows the positional encoding under the standard ICL input sequence. As for the duplicated input of InvICL, we apply the same positional encoding for the original and the repeated examples, as shown in Figure **(b)**.

## A.2    BAG-OF-EXAMPLES ICL

We introduce the implementation detail of two BoE ICL methods mentioned in the main text, PCW (Ratner et al., 2022), SAICL (Cai et al., 2023) and BatchICL (Zhang et al., 2024).

**PCW (Parallel Context Window).** PCW is a work originally aimed at improving the acceptable length of language models. Denote $N$ be the maximal length of a language model, and $n > N$ be the input length. PCW divides the input into context windows with length $N$, and separately puts them into the LM. Finally, it utilizes a "bag-of-window" method (similar to Figure 2(c), where each block in the mask refers to a context window) to generate the predictions. We note that by considering each context example as a window in PCW, it can implement the Bag-of-Examples ICL algorithm.

**SAICL (Structured Attention for ICL).** SAICL is a method proposed to improve the inference efficiency and order-sensitivity of in-context learning. Similar to PCW, they also encode the context examples independently but are also aware of the test example. The original method is based on T5 (Raffel et al., 2020), a language model with the encoder-decoder architecture. We transfer its design to the GPT family by directly taking its attention mask and use the symmetric PE proposed above.

**BatchICL.** Instead of conducting $N$-shot encoding for all context examples, BatchICL utilizes $N$ separate 1-shot encodings for each context example. It then aggregates the intermediate hidden states of the respective last token, which is subsequently incorporated into the forward computation of a zero-shot query to generate the final prediction. We basically follows all the setting introduced in the original paper. As for the layer to extract the aggregated vector, we simply takes the 15-th layer, since they found that any intermediate or later layer is a fair choice.

## A.3    SETTING OF THE EXPERIMENTS ON LINEAR REGRESSION TASKS.

Denote $\mathcal{G} = \{g : \mathcal{X} \in \mathbb{R}^d \to \mathbb{R}, g(\mathbf{x}) = \mathbf{w}^\top \mathbf{x} + b\}$ as the linear function class. Let $\mathcal{D}_{\mathcal{G}}$ be a distribution on $\mathcal{G}$, and $\mathcal{D}_{\mathcal{X}}$ be a distribution on $\mathcal{X}$. In the training phase, we iteratively sample a random function $g \in \mathcal{G}$ from $\mathcal{D}_{\mathcal{G}}$, and sample i.i.d. $\mathbf{x}_1, ..., \mathbf{x}_{k+1}$ from $\mathcal{D}_{\mathcal{X}}$. Then, we produce a prompt in the ICL manner $P = (\mathbf{x}_1, g(\mathbf{x}_1), ..., \mathbf{x}_k, g(\mathbf{x}_k), \mathbf{x}_{k+1})$, and train a model $\theta$ to output $[\hat{g}(\mathbf{x}_1), ..., \hat{g}(\mathbf{x}_k), \hat{g}(\mathbf{x}_{k+1})] = f_\theta(P)$ (as equation Eq. (1)), where $\hat{g}(\mathbf{x}_i)$ is the prediction for $g(\mathbf{x}_i)$.

The training objective is

$$\min_{\theta} \mathbb{E}_{\mathcal{D}_{\mathcal{G}}, \mathcal{D}_{\mathcal{X}}} \left[ \frac{1}{k+1} \sum_{i=0}^{k} \ell(\hat{g}(\mathbf{x}_i), g(\mathbf{x}_i)) \right], \tag{10}$$

where $\ell$ is the MSE loss. In the experiments in Section 4.1, we set $d = 20, k = 40, \mathcal{D}_{\mathcal{X}} = \mathcal{N}(0, I_d)$, and $\mathcal{D}_{\mathcal{G}} : \mathbf{w} \sim \mathcal{N}(0, I_d), b = 0$.

The architecture selection follows (Garg et al., 2022), where a 12-layer GPT-like Transformer decoder is utilized. We implement the four model types by using the symmetric attention mask and PE.

### A.4    IMPLEMENTATION DETAILS OF EXPERIMENTS ON REAL-WORLD DATA.

**Evaluation.**    Following MetaICL (Min et al., 2022b), we consider 7 evaluation settings: 1) HR→LR, which means training with high resource data and testing on low resource data; 2) X→X (X={Classification, QA}), which means training and testing on the same task type, but with no overlap in tasks; 3) Non-X→X (X={Classification, QA, NLI, Paraphrase}, which means training and testing on different task type. The last settings are the most challenging, which require strong generalization capacities of language models (Min et al., 2022b). For each setting, we make evaluations both on all target tasks and a subset that contains target tasks in the unseen domains of the source tasks, e.g., medical, financial, and climate. This setting also challenges the out-of-distribution generalization capability of models.

**Truncation.**    Since MetaICL (Min et al., 2022b) truncates the training sequence when it exceeds the maximum input length of the LM, and the ICL prompt sequence is duplicated in our implementation of InvICL, the training sequences differ between InvICL and other methods because of different truncate rates. As shown in Table 5, there is a significant gap in the dataset size between the standard input and the duplicated input under the truncating setting. To make the comparison fair, we apply the same truncate rate in InvICL to the normal training sequence so that all the methods share the same training set. Additionally, we reduce the number of context examples in the training phase from 16 to 8 to control the truncate rate of InvICL to the same level as standard ICL.

Table 5: Ratio of the remaining data between different input types under the truncating setting of MetaICL (Min et al., 2022b). Here the number of context examples is set to 8.

| INPUT TYPE | HR → LR | CLASS →CLASS | NON-CLASS →CLASS | QA →QA | NON-QA →QA | NON-NLI →NLI | NON-PARA →PARA |
|---|---|---|---|---|---|---|---|
| *Remaining ratio of training dataset* | | | | | | | |
| STANDARD | 70% | 90% | 71% | 59% | 80% | 85% | 85% |
| DUPLICATED | 53% | 79% | 55% | 40% | 62% | 75% | 71% |

**Direct & Channel.**    Besides the standard ICL paradigm, MetaICL (Min et al., 2022b) adopts a new inference paradigm called noisy channel ("Channel") (Min et al., 2022a) and achieves a better experimental performance. Contrary to the standard ICL paradigm (also called "Direct" in (Min et al., 2022b)) that takes $(\mathbf{x}_1, \mathbf{y}_1, ..., \mathbf{x}_n, \mathbf{y}_n, \mathbf{x}_t)$ as input, the Channel paradigm takes $(\mathbf{y}_1, \mathbf{x}_1, ..., \mathbf{y}_n, \mathbf{x}_n, \mathbf{y}_t)$ and try to generate $\mathbf{x}_t$. Note that in order to generate the prediction, Channel ICL needs to perform $n$ forward passes conditioned on each of the $n$ labels $\mathbf{y}_t$ and regard the label with minimum perplexity as the prediction. This will, on the one hand, increase the computational complexity and, on the other hand, reduce its applicability to the generative tasks where the label space is large, *e.g.,* Question Answering. Therefore, we adopt the "Direct" setting in our experiments, *i.e.,* the standard ICL paradigm.

## B    ADDITIONAL EXPERIMENTAL RESULTS

### B.1    SYNTHETIC EXPERIMENTS ON OTHER SETUPS

In this section, we conduct additional synthetic experiments on more functions and out-of-distribution setups, to further showcase the generalization capability of InvICL.

**Other function settings.** We consider two other function settings proposed by (Garg et al., 2022) — sparse linear regression and decision tree, to illustrate the ability of InvICL to learn algorithms to solve other tasks. Results are given in Figure 6.

1. **Sparse linear regression**. In this task, a random linear function $\mathbf{y} = \mathbf{w}^\top \mathbf{x} + b$ is sampled to be predicted, yet the efficient has only 3 non-zero coordinates out of 20 dimensions. Although it is also a linear regression task, its optimal algorithm is no longer least squares but Lasso, which involves solving the least squares objective with an l1-norm regularizer for the weight vector. This demands the in-context learners to learn an algorithm different from that in linear regression to solve this task. Following the experimental settings in our paper, we test the performance of AR ICL and InvICL which are trained with 200k epochs. We can still observe the consistent results of our paper that InvICL possesses fast convergence (InvICL converges while AR ICL does not).

2. **Decision tree**. We follow the setting in (Garg et al., 2022), where the class of depth 4 decision trees with 20-dimensional inputs is considered. We evaluate the performance of AR ICL and InvICL that are trained with 200k epochs. We find that although AR ICL performs better than InvICL for short inputs, as the length of the input sequence increases, InvICL gradually outperforms AR ICL, indicating the strong extrapolation ability of InvICL.

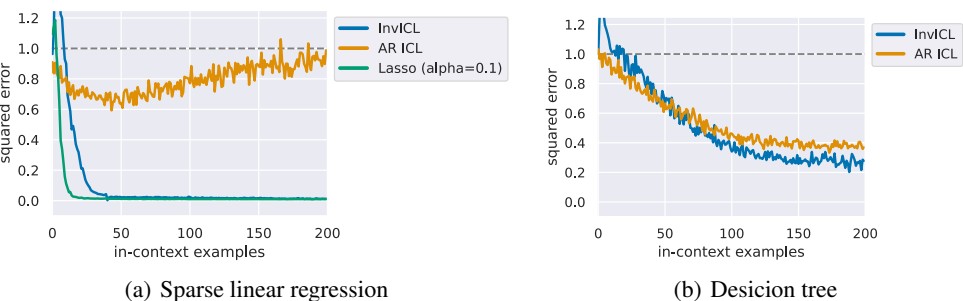

(a) Sparse linear regression        (b) Desicion tree

Figure 6: ICL performance on sparse linear regression and decision tree.

**Out-of-distribution Setups.** We consider three out-of-distribution setups proposed by (Garg et al., 2022; Chen et al., 2023), to showcase the generalization capability of InvICL to out-of-distribution (OOD) tasks. We consider a distribution shift between the training and test datasets. The training data remain consistent with Section A.3. However, for the test data, we apply the following modification:

1. Add **random noise** to the labels by altering $b = 0$ to $b \sim \mathcal{N}(0, 1)$.

2. **Scale** the data sampling by altering $\mathcal{D}_\mathcal{X} = \mathcal{N}(0, I_d)$ to $\mathcal{D}_\mathcal{X} = \mathcal{N}(0, 3^2 I_d)$.

3. Sample the data $x_i$ from a random 10-dimensional **subspace** (out of 20 dimensions).

In Figure 7, we report the testing MSE loss with the models trained for respectively 50k and 200k epochs. We omit Prefix ICL and BoE ICL for their poor performance. We find that InvICL continues the advantages mentioned earlier, *i.e.,* the fast convergence and strong extrapolation ability, indicating its strong capacity on OOD tasks.

## B.2    REAL-WORLD EXPERIMENTS BASED ON GPT-NEO AND PYTHIA

We also conduct experiments with models based on GPT-Neo 2.7B (Black et al., 2021) and Pythia-2.8B (Biderman et al., 2023) with other hyper-parameters unchanged, as shown in Table 6 and 7. The result is similar to what is demonstrated in the main text: InvICL outperforms the baseline in most of the tasks and especially performs well in the OOD settings. This indicates the applicability of InvICL to different base models.

Besides, we note that the three LLMs (GPT-2, GPT-Neo and Pythia) studied in our work utilize three different kinds of PE — trainable PE, Alibi and Rotary PE, respectively. Therefore, our design of symmetric PE is applicable to a wide range of PEs.

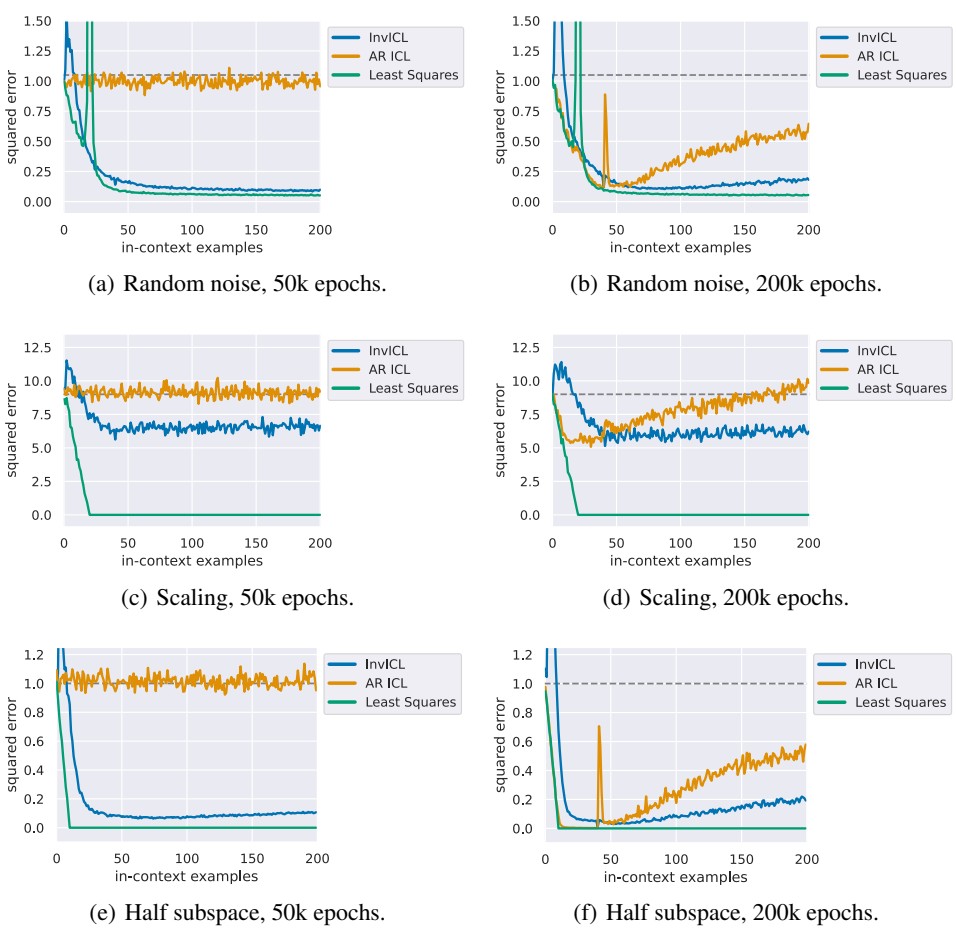

Figure 7: ICL performance on OOD tasks. The training dataset remains consistent with Section 4.1, but we change the distribution of the test dataset. **Random noise**: changing the distribution of the linear bias from $b = 0$ to $b \sim \mathcal{N}(0, 1)$. **Scaling**: changing the sampling distribution of $\mathbf{x}_i$ from $\mathcal{D}_{\mathcal{X}} = \mathcal{N}(0, I_d)$ to $\mathcal{D}_{\mathcal{X}} = \mathcal{N}(0, 3^2 I_d)$. **Half subspace**: Sample the data $x_i$ from a random 10-dimensional subspace (out of 20 dimensions).

Table 6: The in-context learning performance on GPT-Neo 2.7B.

| METHOD | HR → LR | CLASS →CLASS | NON-CLASS →CLASS | QA →QA | NON-QA →QA | NON-NLI →NLI | NON-PARA →PARA | AVG. |
|---|---|---|---|---|---|---|---|---|
| | | | *All target tasks* | | | | | |
| AUTO-REGRESSIVE ICL | 45.8 | **41.2** | 40.1 | 46.4 | **36.8** | **45.2** | 33.1 | 41.2 |
| INVICL(OURS) | **46.1** | 40.2 | **40.2** | **48.6** | 35.8 | 44.7 | **33.7** | **41.3** |
| | | | *Target tasks in unseen domains* | | | | | |
| AUTO-REGRESSIVE ICL | 39.1 | 33.1 | 31.8 | 66.5 | **34.7** | 56.7 | 33.1 | 42.1 |
| INVICL(OURS) | **39.6** | **33.9** | **32.7** | **68.1** | 31.4 | **56.9** | **36.0** | **42.7** |

## B.3 ABLATION STUDY FOR INVICL

In this section, we conduct experiments to test the baselines (AR ICL, PCW, Prefix ICL) using the same duplicated data as InvICL. As shown in Table 8, InvICL still outperforms the baselines when they are given the doubled input as InvICL does.

Table 7: The in-context learning performance on Pythia-2.8B.

| METHOD | HR → LR | CLASS →CLASS | NON-CLASS →CLASS | QA →QA | NON-QA →QA | NON-NLI →NLI | NON-PARA →PARA | AVG. |
|---|---|---|---|---|---|---|---|---|
| | | | *All target tasks* | | | | | |
| AUTO-REGRESSIVE ICL | 31.3 | 22.3 | 27.8 | **33.4** | 33.7 | **29.7** | 37.6 | 30.8 |
| INVICL(OURS) | **31.5** | **26.3** | **28.5** | 33.0 | **35.6** | 28.0 | **40.2** | **31.9** |
| | | | *Target tasks in unseen domains* | | | | | |
| AUTO-REGRESSIVE ICL | 20.8 | 21.0 | 21.0 | 43.5 | 39.7 | **33.5** | 34.2 | 30.5 |
| INVICL(OURS) | **20.9** | **24.2** | **21.1** | **44.6** | **43.7** | **33.5** | **38.6** | **32.4** |

Table 8: Ablation study of using doubling input for the baseline methods. We report the result on HR→LR. InvICL still outperforms the baselines.

| METHOD | DOUBLED INPUT | ORIGINAL INPUT |
|---|---|---|
| AR ICL | 43.8 | 43.4 |
| PCW (BoE ICL) | 40.6 | 39.7 |
| PREFIX ICL | 41.7 | 40.3 |
| INVICL | **45.1** | - |

### B.4 DETAILED RESULTS FOR SYNTHETIC EXPERIMENTS

In this section, we provide detailed results for the synthetic experiments in section 4.1. In figure 8, we demonstrate the error curves of AR ICL and InvICL at different training epochs. In figure 9, we present the error at different training epochs when the number of context examples is 100. Both experiments demonstrate that InvICL's OOD in-context performance (length > 40) consistently outperforms AR ICL across all epochs. Specifically, as shown in figure 9, in the early stages of training, the error of InvICL decreases rapidly, while the error of AR ICL only shows significant reduction after approximately 100k epochs. Furthermore, after 200k epochs, the error of InvICL stabilizes, whereas the error of AR ICL increases.

### B.5 LINEAR PROBING EXPERIMENTS

In this section, we conduct a linear probing experiments based on the synthetic setting, to further explore how the architecture of InvICL impacts the model's internal representations. For a pre-trained model on the synthetic linear regression dataset, we freeze the model parameters and trained a single linear layer on the hidden states of the 3rd, 6th, 9th, and 12th layers, respectively.

As shown in Figure 10, the linear probing error of InvICL is consistent and close to the error curve of the pre-trained model across all tested layers. In contrast, for AR ICL, only the error curve of layer 12 converges to that of the pre-trained model. This indicates that InvICL encodes task features in the model much faster than AR ICL. We believe this is closely related to its context interdependence property, which allows it to utilize richer context example information for encoding.

## C THEORETICAL UNDERSTANDING INVICL FROM AN OPTIMIZATION PERSPECTIVE

In this section, we further characterize the advantages of InvICL from an optimization perspective.

**InvICL Can Approximately Implement Gradient Descent.** Consider a linear regression task with instances $(\mathbf{X}, \mathbf{y})$, where $\mathbf{X}$ consists of row vectors $\mathbf{x}_i^\top \in \mathbb{R}^d$, and $\mathbf{y}$ consists of labels $y_i \in \mathbb{R}$, $i \in [n]$. The goal is to find the optimal weight vector $\mathbf{w}$ that minimizes the LSE loss $\mathcal{L}(\mathbf{w}) = \|\mathbf{X}\mathbf{w} - \mathbf{y}\|^2$. A standard gradient descent (GD) algorithm updates the weights iteratively as follows:

$$\mathbf{w}_\ell = \mathbf{w}_{\ell-1} - \eta \mathbf{X}^\top (\mathbf{X}\mathbf{w}_{\ell-1} - \mathbf{y}), \tag{11}$$

where $\ell$ stands for the iteration step, and $\eta$ is the step size.

Consider the ICL-style model input, formulated as $\mathbf{Z} = (\mathbf{z}_1, ..., \mathbf{z}_n, \mathbf{z}_1, ..., \mathbf{z}_n, \mathbf{z}_t)$, where $\mathbf{z}_j = \begin{pmatrix} \mathbf{x}_j \\ y_j \end{pmatrix}, j \in [n]$ are the context examples, and $\mathbf{z}_t = \begin{pmatrix} \mathbf{x}_t \\ 0 \end{pmatrix}$ is an arbitrary test example. Here we

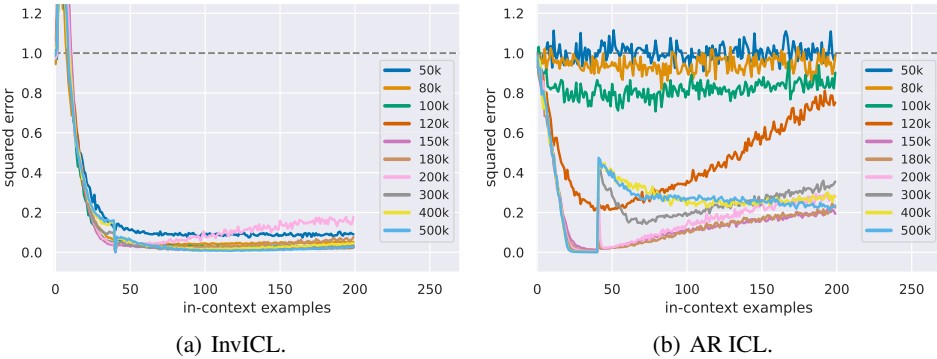

(a) InvICL.              (b) AR ICL.

Figure 8: Intermediate results for InvICL and AR ICL on the linear regression setting. The line colors represent the models trained with different epochs.

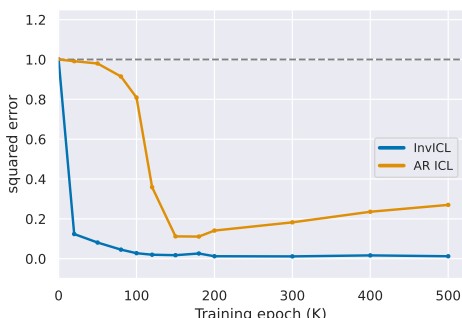

Figure 9: The squared error at different training epochs. We set the number of context examples to 100.

duplicate the input as required by InvICL and expect the model to predict $\begin{pmatrix} \mathbf{x}_t \\ \mathbf{x}_t^\top \mathbf{w} \end{pmatrix}$ at the last token. The theorem below illustrates the evolution of the prediction through the Transformer layer of InvICL.

**Theorem C.1.** *With the attention weight matrices configured as in (Von Oswald et al., 2023), i.e.,*

$$\mathbf{W}_k = \mathbf{W}_q = \begin{pmatrix} \mathbf{I}_{d \times d} & \mathbf{0} \\ \mathbf{0} & 0 \end{pmatrix}, \mathbf{W}_v = \begin{pmatrix} \mathbf{0}_{d \times d} & \mathbf{0} \\ \mathbf{w}_0 & -1 \end{pmatrix}, \mathbf{P} = \eta \mathbf{I}, \tag{12}$$

*InvICL implements the following weight updating rule: at the $\ell$-th layer of the Transformer, the last token outputs $\mathbf{z}_t^{(\ell)} = \begin{pmatrix} \mathbf{x}_t \\ \mathbf{x}_t^\top \mathbf{w}_\ell \end{pmatrix}$, where*

$$\mathbf{w}_\ell = \mathbf{w}_{\ell-1} - \eta \mathbf{X}^\top (\mathbf{X} \mathbf{w}_{\ell-1} - \mathbf{y}) + \eta^2 \Delta \mathbf{w}_{\ell-1}. \tag{13}$$

*Here, $\Delta \mathbf{w}_\ell = \mathbf{X}^\top (\mathbf{X} \mathbf{X}^\top - \text{diag}(\mathbf{X} \mathbf{X}^\top))(\mathbf{X} \mathbf{w}_\ell - \mathbf{y})$.*

Theorem C.1 shows that under specific parametrization, the weight updating rule implemented by InvICL (Eq. (13)) is very similar to that of standard GD (Eq. (11)), differing only by a second-order term. For gradient descent to converge, the step size $\eta$ should be at most the inverse of the largest eigenvalue of $\mathbf{X} \mathbf{X}^\top$. Under this condition, the term $\eta^2 \Delta \mathbf{w}_{\ell-1}$ is dominated by $\eta \mathbf{X}^\top (\mathbf{X} \mathbf{w}_{\ell-1} - \mathbf{y})$, ensuring that InvICL has the potential to closely approximates the standard GD algorithm in this linear regression task.

**Discussion to Other ICL Methods.** We provide a comprehensive comparison of all the ICL methods considered in this paper from the optimization perspective: under the parametrization as in Eq. (12), 1) AR ICL emulates the online GD algorithm (with a constant learning rate) (Ding et al., 2023), which is not guaranteed to converge; 2) Prefix ICL implicitly implements the standard GD algorithm under a specific set of parameters for attention (Von Oswald et al., 2023; Ding et al., 2023); and 3) BoE ICL

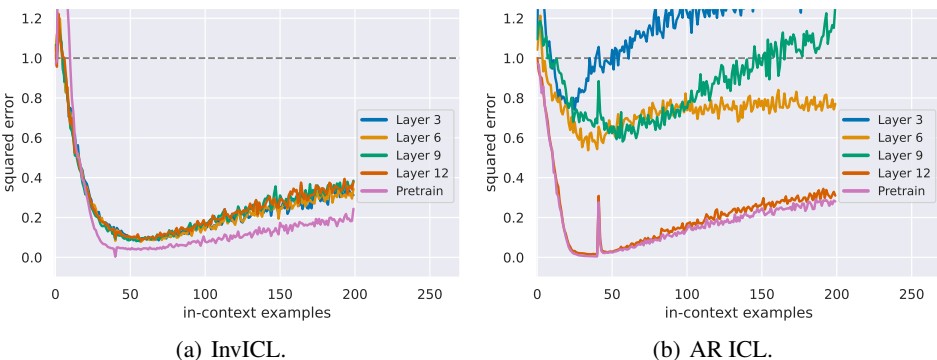

Figure 10: The linear probing results on InvICL and AR ICL.

can only update the weight vector of the test (last) example (not the context examples) without context interdependence. This leads to a constant gradient computed at the initial point, causing it to fail to converge (detailed discussion is in Appendix D.1). Compared with these ICL algorithms, InvICL has several distinct advantages: 1) InvICL surpasses AR ICL in terms of convergence to optimal solutions; 2) Similar to Prefix ICL, InvICL approximately implements the standard GD algorithm while avoiding information leakage; and 3) Unlike BoE ICL, InvICL effectively incorporates context interdependence, allowing it to emulate a more efficient GD algorithm. These advantages underscore the theoretical superiority of InvICL, which synergizes information non-leakage and context interdependence within an invariant ICL framework.

**Practicality of Theorem C.1.** Theorem C.1 is an existence proof which illustrate that the Transformers have the potential to implement complex optimization mechanisms like gradient descent. In fact, The actual weight may not be strictly follow its parametrization. However, empirical studies including Von Oswald et al. (2023); von Oswald et al. (2023), have shown that pre-trained Transformers exhibit behaviors akin to gradient descent in certain scenarios, thereby providing empirical evidence for the theory.

# D    PROOFS

## D.1    PROOF OF THEOREM C.1

*Proof.* We mainly adopt the setting of (Von Oswald et al., 2023) and (Ding et al., 2023). Let $\mathbf{Z} = (\mathbf{z}_1, ..., \mathbf{z}_{2n}, \mathbf{z}_{2n+1}) \in \mathbb{R}^{(d+1)\times(2n+1)}$ be the duplicated input of InvICL, where $\mathbf{z}_j = \begin{pmatrix} \mathbf{x}_j \\ y_j \end{pmatrix}$, $\mathbf{x}_j \in \mathbb{R}^d$, $\mathbf{y}_j \in \mathbb{R}$, and $\mathbf{z}_i = \mathbf{z}_{n+i}$ for $i \in [n]$. Consider the linear self-attention layer in the scheme of InvICL. Given the query, key, value matrix $\mathbf{W}_q, \mathbf{W}_k, \mathbf{W}_v \in \mathbb{R}^{(d+1)\times(d+1)}$ and the projection matrix $\mathbf{P} \in \mathbb{R}^{(d+1)\times(d+1)}$, the updating rule of the layer is as follows:

$$
\begin{aligned}
\mathbf{z}_j &\leftarrow \mathbf{z}_j + \mathbf{P}\mathbf{W}_v\mathbf{z}_j(\mathbf{z}_j^\top \mathbf{W}_k^\top \mathbf{W}_q \mathbf{z}_j), \\
\mathbf{z}_{n+j} &\leftarrow \mathbf{z}_{n+j} + \mathbf{P}\mathbf{W}_v \sum_{i\in[n]\setminus\{j\}} \mathbf{z}_i(\mathbf{z}_i^\top \mathbf{W}_k^\top \mathbf{W}_q \mathbf{z}_{n+j}), \\
\mathbf{z}_{2n+1} &\leftarrow \mathbf{z}_{2n+1} + \mathbf{P}\mathbf{W}_v \sum_{i=1}^{n} \mathbf{z}_{n+i}(\mathbf{z}_{n+i}^\top \mathbf{W}_k^\top \mathbf{W}_q \mathbf{z}_{2n+1}),
\end{aligned}
\tag{14}
$$

where $j \in [n]$. Following the setting of (Von Oswald et al., 2023) and (Ding et al., 2023), we let

$$
\mathbf{W}_k = \mathbf{W}_q = \begin{pmatrix} \mathbf{I}_{d\times d} & \mathbf{0} \\ \mathbf{0} & 0 \end{pmatrix}, \mathbf{W}_v = \begin{pmatrix} \mathbf{0}_{d\times d} & \mathbf{0} \\ \mathbf{w}^{(0)} & -1 \end{pmatrix}, \mathbf{P} = \eta\mathbf{I}.
\tag{15}
$$

Now, we hope to see what kind of iterative algorithm can naturally be implemented by InvICL. Before that, we first give the $L_2^2$ loss after doing one step of gradient descent

$$
\begin{aligned}
&\|\mathbf{X}(\mathbf{w} - \eta \mathbf{X}^\top (\mathbf{X}\mathbf{w} - \mathbf{y})) - \mathbf{y}\|^2 \\
&= \|\mathbf{X}\mathbf{w} - \mathbf{y} - \eta \mathbf{X}\mathbf{X}^\top (\mathbf{X}\mathbf{w} - \mathbf{y})\|^2 \\
&= \|(\mathbf{I} - \eta \mathbf{X}^\top \mathbf{X})(\mathbf{X}\mathbf{w} - \mathbf{y})\|^2.
\end{aligned}
\tag{16}
$$

To compare InvICL with the conventional attention heads for ICL linear regression, here we investigate the convergence properties of the leave-one-out scheme in Eq. (17) viewed as an optimization algorithm for solving the regression problem, and compare it to that of gradient descent. It turns out that if we use the same weighting strategy as (Von Oswald et al., 2023) but with InvICL, then we obtain a similar iterative algorithm for in-context linear regression according to which the last row of $\mathbf{Z}$ evolves, but the update rule transforms into

$$
\mathbf{w}_\ell = \mathbf{w}_{\ell-1} - \eta \mathbf{X}^\top (\mathbf{X}\mathbf{w}_{\ell-1} - \mathbf{y}'),
\tag{17}
$$

where

$$
\mathbf{y}' = \mathbf{y} - \eta \mathbf{X}\mathbf{X}^\top (\mathbf{X}\mathbf{w}_{\ell-1} - \mathbf{y}) + \eta [\mathbf{x}_i^\top \mathbf{x}_i (\mathbf{x}_i^\top \mathbf{w}_{\ell-1} - y_i)]_{i=1}^n
\tag{18}
$$

is the label updated by the leave-one-out scheme. This equation is obtained by first perform a gradient descent step w.r.t. the whole dataset with gradient update $\eta \mathbf{X}^\top (\mathbf{X}\mathbf{w} - \mathbf{y})$ and then minus the term w.r.t the $i$-th data point $\mathbf{x}_i(\mathbf{x}_i^\top \mathbf{w} - y_i)$.

Expanding Eq. (17), we get that the global update becomes

$$
\begin{aligned}
\mathbf{w}_\ell &= \mathbf{w}_{\ell-1} - \eta \mathbf{X}^\top (\mathbf{X}\mathbf{w}_{\ell-1} - \mathbf{y}') \\
&= \mathbf{w}_{\ell-1} - \eta \mathbf{X}^\top (\mathbf{X}\mathbf{w}_{\ell-1} - \mathbf{y} + \eta \mathbf{X}\mathbf{X}^\top (\mathbf{X}\mathbf{w}_{\ell-1} - \mathbf{y}) \\
&\quad - \eta [\mathbf{x}_i^\top \mathbf{x}_i (\mathbf{x}_i^\top \mathbf{w}_{\ell-1} - y_i)]_{i=1}^n) \\
&= \mathbf{w}_{\ell-1} - \eta \mathbf{X}^\top (\mathbf{X}\mathbf{w}_{\ell-1} - \mathbf{y}) + \eta^2 \mathbf{X}^\top \mathbf{X}\mathbf{X}^\top (\mathbf{X}\mathbf{w}_{\ell-1} - \mathbf{y}) \\
&\quad - \eta^2 \mathbf{X}^\top \mathrm{Diag}(\mathbf{X}\mathbf{X}^\top)(\mathbf{X}\mathbf{w}_{\ell-1} - \mathbf{y}).
\end{aligned}
\tag{19}
$$

This delivers Eq. (13). $\qquad \square$

**Remark.** In BoE ICL, since the context examples cannot interact with each other, the GD algorithm implemented by it can only update the weight vector $\mathbf{w}$ of the test (last) example, but not the context examples. Particularly, this means the gradient update process is $\mathbf{w}_\ell = \mathbf{w}_{\ell-1} - g(\mathbf{w}_0, \{\mathbf{x}_i, y_i\})$, where $g$ is the update function of BoE ICL. This means that the gradients are always computed at the initial point of the algorithm, thus the algorithm cannot converge.

### D.2 PROOF OF PROPOSITION 3.4

*Proof.* We will first demonstrate that the attention score matrix $\mathbf{A}$ needs to adhere to a specific form when constrained by the attention mask $\mathbf{M}$, in order to guarantee the permutation equivariance of the embeddings of the context examples. Subsequently, we will establish that this requirement is equivalent to the permutation invariance of the ICL prediction with respect to the context examples.

**Lemma D.1.** *Given an input matrix $\mathbf{H} = (\mathbf{h}_1, ..., \mathbf{h}_n)^\top \in \mathbb{R}^{n \times d}$ and its attention score matrix $\mathbf{A} \in \mathbb{R}^{n \times n}$ defined in Eq. (7). Denote $\mathrm{SA}(\mathbf{H}) = \mathbf{A}\mathbf{H}\mathbf{W}_v\mathbf{P}$ be the self-attention operation, where $\mathbf{A}$ is defined in Eq. (7). Then, $\mathrm{SA}(\mathbf{H})$ is permutation equivariant to $\{\mathbf{h}_i\}$ iff the attention mask $\mathbf{M}$ is equal to*

$$
\begin{pmatrix}
0 & -\infty & \cdots & -\infty \\
-\infty & 0 & \cdots & -\infty \\
\vdots & \vdots & \ddots & \vdots \\
-\infty & -\infty & \cdots & 0
\end{pmatrix},
\begin{pmatrix}
-\infty & 0 & \cdots & 0 \\
0 & -\infty & \cdots & 0 \\
\vdots & \vdots & \ddots & \vdots \\
0 & 0 & \cdots & -\infty
\end{pmatrix}, or\ \mathbf{0}.
$$

*Proof.* Denote $\mathbf{T} \in \mathbb{R}^{n \times n}$ be a permutation matrix on the row vectors of $\mathbf{H}$. This implies that $\mathbf{T} \in \{0, 1\}^{n \times n}$ and $\mathbf{1}_n^\top \mathbf{T} = \mathbf{1}_n^\top$, $\mathbf{T}\mathbf{1}_n = \mathbf{1}_n$. Then the permutation equivariant condition can be

stated as $\mathbf{T}\,\mathrm{SA}(\mathbf{H}) = \mathrm{SA}(\mathbf{TH})$. Since $\mathrm{SA}(\mathbf{H}) = \mathrm{softmax}\left(\mathbf{HW}_q(\mathbf{HW}_k)^\top + \mathbf{M}\right)\mathbf{HW}_v\mathbf{P}$, the condition can be expanded as

$$
\begin{aligned}
&\mathbf{T}\,\mathrm{softmax}\left(\mathbf{HW}_q(\mathbf{HW}_k)^\top + \mathbf{M}\right)\mathbf{HW}_v\mathbf{P} \\
&= \mathrm{softmax}\left(\mathbf{THW}_q\mathbf{W}_k^\top\mathbf{H}^\top\mathbf{T}^\top + \mathbf{M}\right)\mathbf{THW}_v\mathbf{P}.
\end{aligned}
\tag{20}
$$

It can be easily verified that 1) the permutation and softmax operations are commutative, and 2) $\mathbf{T}$ is orthogonal. Therefore, the above equation can be transformed to

$$
\begin{aligned}
&\mathrm{softmax}\left(\mathbf{THW}_q(\mathbf{HW}_k)^\top + \mathbf{TM}\right)\mathbf{HW}_v\mathbf{P} \\
&= \mathrm{softmax}\left(\mathbf{THW}_q\mathbf{W}_k^\top\mathbf{H}^\top + \mathbf{MT}\right)\mathbf{HW}_v\mathbf{P}.
\end{aligned}
\tag{21}
$$

This is equivalent to

$$
\mathbf{TMT}^{-1} = \mathbf{M}
\tag{22}
$$

for arbitrary permutation matrix $\mathbf{T}$. Next, we will discuss what kind of matrix $\mathbf{M}$ satisfies this condition. For notation simplicity, we denote $\mathbf{T}(i,j)$ as the permutation performed only between the $i$-th and $j$-th index.

- Assume $\mathbf{M}_{i,i} = c_1$. Taking $\mathbf{T} = \mathbf{T}(i,j)$, from Eq. (22) we have $\mathbf{M}_{j,j} = c_1$. By iterating over $j$, we have $\mathbf{M}_{k,k} = c_1$ for every $k \in [n]$.

- Assume $\mathbf{M}_{i,j} = c_2, i \neq j$. Taking $\mathbf{T} = \mathbf{T}(i,k), k \neq j$, from Eq. (22) we have $\mathbf{M}_{k,j} = c_2$; taking $\mathbf{T} = \mathbf{T}(j,k), k \neq i$, we have $\mathbf{M}_{i,k} = c_2$. Hence, by iterative applying permutations in this way, we can conclude that $\mathbf{M}_{k,l} = c_2$ for every $k \neq l$.

In conclusion, $\mathbf{M} = c_1\mathbf{I}_n + c_2(\mathbf{1}_{n\times n} - \mathbf{I}_n)$. Since the elements of an attention mask can only take the value of either $0$ or $-\infty$, $\mathbf{M}$ can only be one of the three forms demonstrated in Lemma D.1 (an all $-\infty$ attention mask is meaningless). $\qquad\square$

Now we prove the equivalence between the desired permutation invariance property and the equivariance property discussed in Lemma D.1. As the permutation invariance property involves the ICL prediction, which relies on the test embedding $\mathbf{h}_t$, it is necessary to incorporate it into the discussion. We denote the full input matrix of ICL as $\tilde{\mathbf{H}} = (\mathbf{h}_1, ..., \mathbf{h}_n, \mathbf{h}_t) \in \mathbb{R}^{(n+1)\times d}$, and the corresponding matrices in the self-attention process as $\tilde{\mathbf{A}}, \tilde{\mathbf{M}}$.

**Lemma D.2.** *Let the output embeddings of the Transformer be* $\mathbf{H}' = (\mathbf{h}'_1, ..., \mathbf{h}'_n, \mathbf{h}'_t)$. *Then,* $\mathbf{h}'_t$ *is* **invariant** *to the permutation of* $(\mathbf{h}_1, ..., \mathbf{h}_n)$ *iff* $(\mathbf{h}'_1, ..., \mathbf{h}'_n)$ *is* **equivariant** *to the permutation of* $(\mathbf{h}_1, ..., \mathbf{h}_n)$.

*Proof.* First, we need to extend existing results to the circumstance of the full input $\tilde{\mathbf{H}}$. Consider the attention mask $\tilde{\mathbf{M}} \in \mathbb{R}^{(n+1)\times(n+1)}$ of the full input, whose $n \times n$ submatrix at the upper-left is equal to $\mathbf{M}$, *i.e.,* $\tilde{\mathbf{M}}_{1:n,1:n} = \mathbf{M}$. From the condition in the Proposition we have that $\tilde{\mathbf{M}}_{n+1,:} = \mathbf{0}_{n+1}^\top$. Besides, it is evident that Proposition 3.5 also satisfies for $\tilde{\mathbf{M}}$, we have $\tilde{\mathbf{M}}_{1:n,n+1} = -\infty \cdot \mathbf{1}_n^\top$. Other variables can be naturally extended.

In Lemma D.1, we have proved that the equivariance property is equivalent to the attention mask $\mathbf{M}$ being one of three specific forms. Now we prove the contrapositive statement of Lemma D.2.

If $(\mathbf{h}'_1, ..., \mathbf{h}'_n)$ is not equivariant to the permutation of $(\mathbf{h}_1, ..., \mathbf{h}_n)$, by Lemma D.1, the mask on context examples $\mathbf{M}$ must satisfy either **1)** $\exists i \neq j, \mathbf{M}_{ii} \neq \mathbf{M}_{jj}$, or **2)** $\exists i \neq j, k \neq l, \mathbf{M}_{ij} \neq \mathbf{M}_{kl}$. We separately demonstrate that these properties will break the property of permutation invariance. For the following circumstances, we uniformly let $\mathbf{W}_q = \mathbf{W}_k = \mathbf{W}_v = \mathbf{P} = \mathbf{I}_{n+1}$. Denote the embedding of $\mathbf{h}_i$ after $k$ self-attention layer as $\mathbf{h}_i^{(k)}$. Then, the embeddings are updated as

$$
\mathbf{h}_i^{(k+1)} = \sum_{j=1,...,n,t} [s(\mathbf{h}_i^{(k)}, \mathbf{h}_j^{(k)}) + \tilde{\mathbf{M}}_{ij}]\mathbf{h}_j^{(k)},
\tag{23}
$$

where $s(\cdot, \cdot)$ is the similarity function calculated by their inner product and softmax normalization, which is defined in 2.

- $\exists i \neq j, \mathbf{M}_{ii} \neq \mathbf{M}_{jj}$. Without loss of generality, since the elements of $\mathbf{M}$ only take the value of either $0$ or $infty$, we let $\mathbf{M}_{11} = 0, \mathbf{M}_{22} = -\infty$. Then we construct the input matrix as $\mathbf{h}_1 = \mathbf{e}_1, \mathbf{h}_2 = \mathbf{e}_2, \mathbf{h}_i = \mathbf{0}(i > 2), \mathbf{h}_t = \mathbf{0}$, where $\mathbf{e}_i$ denotes the $i$-th unit vector ($i \in [d]$). Since $\mathbf{M}_{22} = \mathbf{M}_{2,n+1} = -\infty$, following Eq. (23), we find that $\mathbf{h}_2^{(1)} = c_1\mathbf{e}_1$. And since $\mathbf{M}_{11} = 0$, we have $\mathbf{h}_1^{(1)} = c_2\mathbf{e}_1 + c_3\mathbf{e}_1$.

  Now we permute the first and second examples, *i.e.,* $\mathbf{h}_1 = \mathbf{e}_2, \mathbf{h}_2 = \mathbf{e}_1$. Although we find that the first update of the test embedding remains unchanged since Eq. (23) is permutation invariant for $\mathbf{h}_t^k$, the second update differs. Since we have $\mathbf{h}_2^{(1)} = c_1\mathbf{e}_2$ and $\mathbf{h}_1^{(1)} = c_3\mathbf{e}_1 + c_2\mathbf{e}_1$, the aggregation $\mathbf{h}_i^{(2)}$ changes. Therefore, the property of permutation invariance is broken.

- $\exists i \neq j, k \neq l, \mathbf{M}_{ij} \neq \mathbf{M}_{kl}$. Without loss of generality, let $\mathbf{M}_{ij} = 0, \mathbf{M}_{kl} = -\infty$. We construct $\mathbf{h}_i = \mathbf{e}_1, \mathbf{h}_k = \mathbf{e}_2, \mathbf{h}_{\neq i,k} = \mathbf{0}$. Then, we have $\mathbf{h}_j^{(1)} = c_1\mathbf{e}_1 + c_2\mathbf{e}_2$, and $\mathbf{h}_l^{(1)} = c_3\mathbf{e}_1$. Similar to the above process, we can prove that $\mathbf{h}_t$ is not permutation invariant w.r.t. the index exchange $(i,j) \leftrightarrow (k,l)$.

In conclusion, any attention mask $\mathbf{M}$ that violates Lemma D.1 will break the property of permutation invariance. Thus Lemma D.2 is proved. $\square$

Finally, by combining Lemmas D.1 and D.2, we can deliver Proposition 3.4. $\square$

### D.3 Proof of Proposition 3.5

*Proof.* Consider the case that $G$ has no self-loops. Since $G$ is a digraph with no cycles, it is a directed acyclic graph (DAG). According to the graph theory (Cormen et al., 2022), DAG can be topologically ordered, which means in this ordering, any vertex is not reachable from later vertices in the graph. Therefore, if we reorder the vertices in this way, we have $\mathbf{A}_{ij} = 0$ for $i \leq j$, which infers that $\mathbf{A}$ is strictly lower diagonal. Since the original graph allows self-loop, which corresponds to the diagonal elements, the adjacency matrix is lower triangular. This is equivalent to that the attention mask on context examples $\mathbf{M}$ is lower triangular. $\square$

## E  RELATED WORK

**The order-sensitivity of ICL.** The phenomenon that ICL is sensitive to the permutation of context examples has been observed in several works. (Zhao et al., 2021) and (Lu et al., 2022) used GPT-3 to perform in-context learning on classification tasks such as SST-2 and observe a high variance w.r.t. the permutation of the context examples. Besides, (Xie et al., 2021) and (Agrawal et al., 2022) found the same phenomenon on a generated synthetic dataset and machine learning tasks, respectively. Additionally, (Chen et al., 2022) empirically showed that the order-sensitivity is negatively correlated to the performance of ICL. To address this issue, (Zhao et al., 2021) proposed a calibration module to make the output distribution consistent with prior knowledge. (Lu et al., 2022) filtered out the best prompt ordering by investigating their calibration on a generated set. (Xiang et al., 2024) utilizes contrastive learning to align representations of in-context examples across different positions, resulting in the alleviation of order sensitivity. Besides, there are works that focuses on implementing the concept of permutation invariance from an architectural perspective. For example, SAICL (Cai et al., 2023) proposed a structured attention mechanism that achieves permutation invariance. However, their work is based on improving the ICL performance of T5 (Raffel et al., 2020), a language model based on an encoder-decoder architecture, which did not address the order-sensitivity issue of auto-regressive LMs. BatchICL (Zhang et al., 2024) is the work that is most relevant to us. Instead of conducting $N$-shot encoding for all context examples, it utilizes $N$ separate 1-shot encodings for each context example. It then aggregates the intermediate hidden states of the respective last token, which is subsequently incorporated into the forward computation of a zero-shot query to generate the final prediction. In this way, the model achieves permutation invariance since the encoding of the context examples are independent.

**The connection between ICL and Gradient Descent.** Early stage formal theoretical investigation of the linear regression in-context learners includes (Akyürek et al., 2022; Von Oswald et al., 2023). They first showed that Transformers learn in context via gradient descent, where one layer performs

one gradient update. In subsequent work, (von Oswald et al., 2023) further argued that Transformers are strongly biased towards learning to implement gradient-based optimization routines. (Ahn et al., 2023) showed Transformers can learn to implement preconditioned Gradient Descent, where the pre-conditioner can adapt to the data. (Bai et al., 2023) provided detailed constructions for how Transformers can implement a range of learning algorithms via gradient descent. (Dai et al., 2022) conducted experiments on NLP tasks and concluded that Transformer-based language models performing ICL behave similarly to models fine-tuned via gradient descent; however, concurrent work argued that real-world LLMs do not perform ICL via gradient descent (Shen et al., 2023). (Fu et al., 2023) argued that Transformers actually learn to perform in-context learning by implementing a higher-order optimization method, not gradient descent. Predictions made by different Transformer layers match iterations of higher-order optimization methods better than they match iterations of gradient descent.

