# OpenReview forum: "Rethinking Invariance in In-context Learning"
_ICLR.cc/2025/Conference — ICLR 2025 Poster_

### Official Review · Reviewer_LHUi · 2024-10-29

**Soundness:** 4
**Presentation:** 4
**Contribution:** 3
**Rating:** 6
**Confidence:** 3

**Summary:**

This paper proposes InvICL, a new invariant ICL algorithm that satisfies not only invariance, but also information non-leakage and context interdependence. Then the authors provide both theoretical and empirical evidence to prove the effectiveness of the proposed algorithm on ICL.

**Strengths:**

- The paper is overall well written and well presented.
- The paper present reasonable desiderata for ICL algorithms, and the proposed method complements prior works on those criteria.
- The paper present some theoretical intuition on why the proposed method should work better than its counter parts.
- Authors conduct various experiments to prove the effectiveness of the proposed algorithm.

**Weaknesses:**

- One might view the proposed method as concatenation of the prior works (PrefixICL and (variant of) PCW)
- As with the previous works, it lacks connection with *actual* ICL performed by LLMs.
- Although the inference time of InvICL does not differ much from AR ICL, is it also true for training? To my understanding, in InvICL (and other invariant ICL algorithms), $\hat{g}(x_i)$ and $\hat{g}(x_{i+1})$ cannot be computed on one forward pass.
- For real-world dataset experiments, is AR ICL also fine-tuned on ICL tasks? If not, comparing ICL specific fine-tuned model and general LLM on ICL tasks seems unfair.

**Questions:**

- Even with InvICL, is there a *learning plateau* during training?
- If we actually train with InvICL, does the weight matrices aligns with Theorem 4.1?
- Why is adding only symmetric positional embedding not beneficial?

---

> ### Author Response · Authors · 2024-11-24
> **Response to reviewer LHUi**
>
> We thank reviewer LHUi for appreciating the presentation and solidness of our paper. Now we address your concerns below.
>
> ---
>
> **Q1.** One might view the proposed method as concatenation of the prior works (PrefixICL and (variant of) PCW)
>
> **A1.** We respectfully disagree that this is a simple combination of existing techniques. As we stated in Proposition 3.6, if we simply combine the characteristics of PrefixICL (context interdependence) and PCW (information non-leakage), the corresponding attention mask **must be diagonal**, which clearly does not work. Consequently, we use a **leave-one-out two-step encoding technique** to accommodate these two properties into the invariant ICL algorithm simultaneously, which is technically different from existing methods like Prefix ICL and PCW.
>
> ---
>
> **Q2.** As with the previous works, it lacks connection with *actual* ICL performed by LLMs.
>
> **A2.** We guess that you might be referring to our fine-tuning process as not aligning with the ICL paradigm. However, after the fine-tuning process to enhance the capability of invariant ICL, its usage still adheres to the standard ICL paradigm.
>
> ---
>
> **Q3.** Although the inference time of InvICL does not differ much from AR ICL, is it also true for training? To my understanding, in InvICL (and other invariant ICL algorithms), g(xi) and g(xi+1)  cannot be computed on one forward pass.
>
> **A3.** As stated in line 402, we compute gradients only on the label of the test example, so no multiple forward passes are required. Moreover, we truncated the doubled input for InvICL during the training phase to match the input length of AR ICL (as discussed in Appendix A.4), resulting in no difference in training time costs between these methods in this codebase (approximately 200 minutes on 8 Nvidia A100 GPUs).
>
> ---
>
> **Q4.** For real-world dataset experiments, is AR ICL also fine-tuned on ICL tasks? If not, comparing ICL specific fine-tuned model and general LLM on ICL tasks seems unfair.
>
> **A4.** Yes, we applied the same training process to all ICL variants (including AR ICL), which directly follows the setup of MetaICL without additional hyperparameter tuning.
>
> ---
>
> **Q5.** Even with InvICL, is there a *learning plateau* during training?
>
> **A5.** In Figure 9, we plotted the squared error curves of AR ICL and InvICL over training steps in simulation experiments. The results show that InvICL achieves relatively low error right from the beginning of training, while AR ICL only starts to decrease in error after nearly 100K epochs. This result indicates that AR ICL exhibits a learning plateau, whereas InvICL does not, which is an advantage of InvICL. We have included the above results in Appendix B.4.
>
> ---
>
> **Q6.** If we actually train with InvICL, does the weight matrices aligns with Theorem 4.1?
>
> **A6.** The actual weight may not be strictly follow the parametrization in Theorem 4.1. Following the research approach of a series of previous works, Theorem 4.1 is an **existence proof** to explore the theoretical property of InvICL. It illustrates that the architecture of InvICL has the ability to implement gradient descent, rather than it should align with the weights.
>
> In fact, there are recent work that validates Transformers simulating gradient descent from the perspective of training dynamics [1, 2]. And we will also consider exploring the connection between InvICL’s training dynamics and gradient descent in future work.
>
> [1] von Oswald, Johannes, et al. "Transformers learn in-context by gradient descent." *arXiv preprint arXiv:2212.07677* (2022).
>
> [2] Von Oswald, Johannes, et al. "Uncovering mesa-optimization algorithms in transformers." *arXiv preprint arXiv:2309.05858* (2023).
>
> ---
>
> **Q7.** Why is adding only symmetric positional embedding not beneficial?
>
> **A7.** This is a good question. We guess it is because the primary factor preventing AR ICL from achieving permutation invariance lies in the asymmetric causal mask, rather than the asymmetric positional encoding (PE). As shown in Figure 7 (cited below), when only symmetric PE is added, the model’s sensitivity to permutations actually increases, leading to a drop in performance. In contrast, when only the invariant mask is applied, the model’s sensitivity significantly decreases, resulting in improved performance. This suggests that symmetric PE can only serve as a supplementary component to the invariant mask and cannot be used independently.
>
> ---
>
> Thanks again for you detailed review. Please let us know if there is more to clarify.

---

> ### Comment · Reviewer_LHUi · 2024-11-25
> **Response to the feedback**
>
> Thank you for the detailed response. I have some comments to make, and feel free to response if the time allows (the score will not depend on whether or not authors respond to the following comments).
>
> **Q3**: If this is the case, then the experimental detail in Appendix A.3 and line 406 (in revised version) seem to be contracting one another. Nevertheless, the strongest advantage of Transformer and casual mask is that during training you can compute the whole sequence at once. However, with the proposed mask (and the other invariant ICL ones), I believe the shape of mask is different for different sequence length. This, in my opinion, is the major drawback of this work.
>
> **Q5**: It is very surprising that changing the mask structure can significantly reduce the learning plateau of ICL. This might help to identify the cause of the plateau. However, the increase in training loss of AR ICL seems bizarre, because it has never been reported in the prior works.
>
> My opinion on this work is overall positive. However, I think my current score is an appropriate assessment.

---

> ### Author Response · Authors · 2024-11-25
>
> Thanks for your feedback. We address your additional concerns below.
>
> **Q3.** Appendix A.3 describes the setting of the synthetic tasks, while line 406 (after another revised version following the suggestion of reviewer k8s4, it becomes line 350) describes the setting of real-world task. Thus, there is no contradiction. In fact,
> for the synthetic tasks, we indeed perform a single forward pass on the sequence and compute the loss over all label tokens. Although technically, predicting the previous context examples is not InvICL, we find that this training process still yields a good InvICL performance. This also suggests that InvICL is sufficiently powerful and adaptable to different training paradigms.
>
> **Q5.** The y-axis in figure 9 is not training error but test error. The results are obtained by testing sequences with 100 context examples, whereas during training, the maximum number of context examples is 40. Therefore, the increase in AR ICL error indicates its tendency to overfit the training sequences, while the stability of InvICL error demonstrates its strong length generalization capability.
>
> We hope this addresses your concerns, and we are happy to answer any further questions if additional clarification is needed.

---

### Official Review · Reviewer_fSCp · 2024-11-02

**Soundness:** 4
**Presentation:** 4
**Contribution:** 3
**Rating:** 6
**Confidence:** 4

**Summary:**

The paper presents a new method to achieve higher performance out of In-Context Learning with no additional model training. It presents conditions under which a well defined notion of invariant In-context learning should perform well highlighting the gap in prior works that this paper fills with its proposed method. The paper gives theoretical and empirical justification for the superiority of its method.

**Strengths:**

- The paper presents a clear step up in terms of theoretical ideas as well as empirical evidence of improvement compared to prior works attempting to optimize ICL.
- Paper is very well written, clearly presents and distinguishes its contribution.
- Although the method requires more computation in theory, the authors achieve parallelism and same order of computation as standard ICL with a smart trick. The authors also talk about the additional memory requirements. [Point being that the paper addresses a plethora of relevant points surrounding its method].

**Weaknesses:**

**The significance of section 4 and section 5.1 are unclear.** Please see this [ICML 2024 paper](https://arxiv.org/abs/2310.08540) that talks about how training transformers with ICL objective may be incompatible with real ICL in LLMs that do not train explicitly for ICL with fixed ICL prompt format.
- Theorem 4.1 shows that if we put weight matrices in a particular format, we can simulate InvICL with transformers. But, is there reason to believe that trained transformers end up with similar weights?
- Similarly, results from section 5.1 assume that transformers are trained with ICL objective, which is not true for LLMs.

I believe these points merit a discussion in the paper.

**Questions:**

- **Definition 3.3**: if j > i, and this condition still holds, then isn’t it sort of contradicting information non-leakage (not technically with your definition as it only requires non-leakage to the corresponding label; but in my opinion, if an x_i can influence another x_i or y_i later in the sequence, it should count as information leakage)? Should the condition be j < i instead of j \neq i? This is intuitively the case for AR LLMs because they only depend on previous context and you have mentioned it in point 3 (“These examples provide the context for better encoding of x_i, which in turn improves the prediction of future examples when x_i serves as their context.”) Why is context interdependence important, or why should I believe that context interdependence should be useful/necessary for AR LLMs, when they can not attend to future demonstrations?
- External modification of attention masks (through aggregation via BoE) is a deviation from how these models are trained to read those activations. How do you think this impacts the model’s internal representations? What could be the effects on predictions apart from pure performance numbers (like does it increase hallucination)? Would like to see some discussion or simple experiment along this line.

PS: I hope to increase my rating if the authors make an effort to discuss my concerns.

---

> ### Author Response · Authors · 2024-11-24
> **Response to reviewer fSCp (Part 1/2)**
>
> We thank reviewer fSCp for acknowledging the solidness of our work. We will address your main concerns in the following points.
>
> ---
>
> **Q1. The significance of section 4 and section 5.1 are unclear.** Please see this [ICML 2024 paper](https://arxiv.org/abs/2310.08540) that talks about how training transformers with ICL objective may be incompatible with real ICL in LLMs that do not train explicitly for ICL with fixed ICL prompt format.
>
> - Theorem 4.1 shows that if we put weight matrices in a particular format, we can simulate InvICL with transformers. But, is there reason to believe that trained transformers end up with similar weights?
> - Similarly, results from section 5.1 assume that transformers are trained with ICL objective, which is not true for LLMs.
>
> I believe these points merit a discussion in the paper.
>
> **A1.** We appreciate the reviewer’s observation. We address your concerns about section 4 and section 5.1 respectively.
>
> - **Section 4**. Theorem 4.1 is an **existence proof**. It illustrates that the Transformer architecture has the ability to implement gradient descent, rather than suggesting that a pre-trained Transformer necessarily adheres to this parameterization. In fact, empirical studies including [1, 2], have shown that pre-trained Transformers exhibit behaviors akin to gradient descent in certain scenarios, thereby providing empirical evidence for the theory.
> - **Section 5.1.**  Our paper focuses on improving the ICL capability of LLMs, rather than investigating the reasons behind the emergence of ICL ability. Therefore, we train the model using the ICL objective to demonstrate that InvICL can achieve stronger ICL capability compared to traditional AR ICL. This is also aligned with the objective we use in the real-world experiments (section 5.2).
>
> Thanks again for your constructive suggestion, and we have added the discussion in Section 6.
>
> [1] von Oswald, Johannes, et al. "Transformers learn in-context by gradient descent." *arXiv preprint arXiv:2212.07677* (2022).
>
> [2] Von Oswald, Johannes, et al. "Uncovering mesa-optimization algorithms in transformers." *arXiv preprint arXiv:2309.05858* (2023).
>
> ---
>
> **Q2.** Definition 3.3: if j > i, and this condition still holds, then isn’t it sort of contradicting information non-leakage (not technically with your definition as it only requires non-leakage to the corresponding label; but in my opinion, if an x_i can influence another x_i or y_i later in the sequence, it should count as information leakage)? Should the condition be j < i instead of j \neq i? This is intuitively the case for AR LLMs because they only depend on previous context and you have mentioned it in point 3 (“These examples provide the context for better encoding of x_i, which in turn improves the prediction of future examples when x_i serves as their context.”) Why is context interdependence important, or why should I believe that context interdependence should be useful/necessary for AR LLMs, when they can not attend to future demonstrations?
>
> **A2.** Definition 3.3 does not conflict with the definition of information non-leakage. In fact, we highlight information non-leakage in order to ensure that there are no shortcuts in the encoding process. Thus, we only require that the sample $x_i$ cannot see its label $y_i $, without restricting it from accessing other context examples, as they do not contain information about $y_i$. Consequently, it is sufficient to require that  $i \neq j$  in the both Definition 3.2 and 3.3.
>
> As for the importance of context interdependence, it mainly comes from the findings in our experiments that **even being invariant, BoE ICL (PCW) is generally much inferior to AR ICL in practice**. As shown in Figure 1 (motivation experiment) and Table 2 (main experiment), AR ICL outperforms BoE ICL on almost all tasks by a large margin. Examining their differences, we observe that compared to BoE ICL, a key advantage of AR ICL is that each training example is encoding with the knowledge of previous examples, so the ICL ability gradually increases as $N$ increases. Instead, in BoE ICL, encodings are independent and each example is treated as the first example. Therefore, the key drawback of BoE ICL is that it cannot leverage the growing number of other examples to improve the encoding of each training example. This is why we bring “context interdependence” as a key desiderata of ICL and define it in this way.
>
> ---

---

> ### Author Response · Authors · 2024-11-24
> **Response to reviewer fSCp (Part 2/2)**
>
> **Q3.** External modification of attention masks (through aggregation via BoE) is a deviation from how these models are trained to read those activations. How do you think this impacts the model’s internal representations? What could be the effects on predictions apart from pure performance numbers (like does it increase hallucination)? Would like to see some discussion or simple experiment along this line.
>
> **A3.** This is an interesting question. To further explore how the architecture of InvICL impacts the model’s internal representations, we conducted additional probing experiments in Appendix B.5. For a pre-trained model on the synthetic linear regression dataset, we froze the model parameters and trained a single linear layer on the hidden states of the 3rd, 6th, 9th, and 12th (last) layers, respectively. Since the only comparable model to InvICL is AR ICL, we conducted the experiment on these two models.
>
> As shown in Figure 10, the linear probing error of InvICL is consistent and close to the error curve of the pre-trained model across all tested layers. In contrast, for AR ICL, only the error curve of layer 12 converges to that of the pre-trained model. This indicates that InvICL encodes task features in the model much faster than AR ICL. We believe this is closely related to its context interdependence property, which allows it to utilize richer context example information for encoding.
>
> ---
>
> Thanks again for your thoughtful questions. Please let us know if there is more to clarify, and we are happy to take your further question in the discussion stage.

---

> > ### Comment · Reviewer_fSCp · 2024-11-25
> >
> > A1. I have read von Oswald et al., and other related papers, and already know that they show **existence** of weights that can simulate gradient descent in transformers. Also, the empirical evidence they show is when these transformers are trained with _ICL objective_, which is incompatible with real LLMs. This evidence does not show anything for real LLMs where the training distribution is very different from the prompt distribution used to elicit ICL. That is why I pointed to the ICML 2024 paper, which talks about this. Section 4 is just repeating the same thing these previous works showed. I agree with reviewer k8s4 and suggest that it be replaced. Similarly, section 5.1 shows experiments that use _ICL objective_ training similar to von Oswald et al., and do not reveal the nature of real LLMs. On the other hand, section 5.2 shows the real application of this method which is good. I would reduce emphasis on these two sections as the presented method provides a much more practical benefit and does not benefit from connecting with previous incompatible theory.
> >
> > A2. Agree about shortcuts. However, the statement "... without restricting it from accessing other context examples, as they do not contain information about $y_i$" does not paint the whole picture. [Task recall](https://arxiv.org/abs/2305.09731) can work even with [random labels](https://arxiv.org/pdf/2202.12837), so there might be some information leakage even with just other context examples. Somewhat convinced about the usefulness of context-interdependence, though still feel there might be some incompatibility with non-leakage.
> >
> > A3. Appreciate the additional experiments.
> >
> > I will keep my score, but will vote for accepting this paper at the conference.

---

> > > ### Author Response · Authors · 2024-11-25
> > >
> > > Thanks again for your feedback and nominating our paper for acceptance. We have just updated the manuscript to move section 4 to appendix.
> > >
> > > **About A2**: As you mentioned in the related work [1], accessing random labels from other samples helps the model improving abilities in task call (also called "task recognition"). By understanding the task represented by the input sequence, the model would effectively improve in the predictions of ICL. This further emphasizes that this part of the information interaction is helpful and should be preserved rather than considered as information leakage and removed.  In contrast, accessing the information of the tokens' own labels harms the model’s ICL capabilities, as the labels are not available during testing. This is why we consider only this as information leakage.
> > >
> > > [1] Pan, Jane, et al. "What In-Context Learning" Learns" In-Context: Disentangling Task Recognition and Task Learning." arXiv preprint arXiv:2305.09731 (2023).

---

### Official Review · Reviewer_p8nF · 2024-11-03

**Soundness:** 3
**Presentation:** 2
**Contribution:** 3
**Rating:** 6
**Confidence:** 3

**Summary:**

This paper presents InvICL, a new approach to in-context learning that achieves three key properties: permutation invariance, information non-leakage, and context interdependence. The authors propose a parallel implementation using a duplicated input sequence and modified attention patterns. They provide a theoretical analysis showing that InvICL approximates standard gradient descent and demonstrates improved performance over baseline methods (GPT-2 Large 762M, GPT-Neo 2.7B, Pythia-2.8B). The work tries to understand and improve in-context learning, though there are some practical concerns.

**Strengths:**

- The paper identifies and formalizes three important properties for ICL that weren't previously unified. The authors demonstrate why these properties matter.
- The theoretical analysis is simple but straightforward. The authors prove that InvICL approximates standard gradient descent (Theorem 4.1) and show how this leads to better convergence properties compared to other ICL variants.
- The experimental results look interesting. The method shows strong performance across multiple settings - synthetic tasks (Figure 3), out-of-distribution scenarios (Figure 7), and real-world datasets (Table 2).

**Weaknesses:**

- The practical applicability of the method raises some concerns. The paper relies on MetaICL finetuning, which is computationally expensive for modern large language models. I wonder if there are any training-free methods.
- The efficiency implications are concerning. Doubling the input sequence length (as shown in Figure 2d) increases memory usage. In Section 5.2, “We find that when the inputs size of the GPT-2 Large model increases from 512 to 1024, the GPU memory overhead increases by 14% (from 4.2 GB to 4.8GB)”. However, when the context is long, e.g., 64k, the overhead will be super large. Moreover, the attention pattern used likely makes it incompatible with FlashAttention optimizations, which could further impact practical deployment.
- The experimental setup feels somewhat dated by using GPT-2 Large as the primary model. While the authors include some results with GPT-Neo 2.7B and Pythia-2.8B in the appendix, evaluating more recent models like Phi-3.5 or Llama 3.2 would better demonstrate the method's relevance to current architectures.
- There are some inconsistencies in the presentation. Figure 1 shows results for PCW while Table 1 uses different terminology (Inv ICL), making it difficult to track the method comparisons. It confuses me to understand the relative performance of different approaches.

**Questions:**

How would InvICL perform in few-shot settings without MetaICL finetuning? This would be particularly relevant for scenarios where finetuning isn't practical.

---

> ### Author Response · Authors · 2024-11-24
> **Response to reviewer p8nF**
>
> We thank reviewer p8nF for acknowledging both theoretical and empirical results of our work. We now address your concerns in the following points.
>
> ---
>
> **Q1.** The practical applicability of the method raises some concerns. The paper relies on MetaICL finetuning, which is computationally expensive for modern large language models. I wonder if there are any training-free methods.
>
> **A1.** Due to the asymmetric characteristics of autoregressive Transformers, achieving invariance requires structural modifications. However, directly altering the structure often leads to performance degradation. One training-free invariant ICL approach is to consider all possible permutations of the n context examples, perform inference for each permutation, and then aggregate the results. However, it is obvious that the complexity of this algorithm is extremely high (requiring n! inferences each time).
>
> Considering these factors, we believe that enhancing the performance of invariant ICL methods in a wide range of ICL scenarios through a short fine-tuning process (~200min training with 8 Nvidia A100 GPUs) is feasible.
>
> ---
>
> **Q2.** The efficiency implications are concerning. Doubling the input sequence length (as shown in Figure 2d) increases memory usage. In Section 5.2, “We find that when the inputs size of the GPT-2 Large model increases from 512 to 1024, the GPU memory overhead increases by 14% (from 4.2 GB to 4.8GB)”. However, when the context is long, e.g., 64k, the overhead will be super large. Moreover, the attention pattern used likely makes it incompatible with FlashAttention optimizations, which could further impact practical deployment.
>
> **A2.** Although InvICL doubles the input length, the theoretical memory overhead is not as large as expected due to the sparsity of its mask. Theoretically, the number of attention computations required by InvICL is on the same order of magnitude as full attention, and only about twice that of causal attention (we added detailed analysis of the complexity in section 6).
>
> Although this theoretical space complexity currently lacks support in the FlashAttention implementation, InvICL can be realized through a taylor-down approach at the CUDA kernel level. In fact, exploring efficient implementations of attention mechanisms beyond the standard attention remains an active area of research [1, 2]. Therefore, we believe that implementing InvICL to achieve its theoretical complexity is feasible and are willing to explore it further in the future.
>
> [1] Pytorch, FlexAttention. https://pytorch.org/blog/flexattention/
>
> [2] flashattention2-custom-mask. https://github.com/alexzhang13/flashattention2-custom-mask
>
> ---
>
> **Q3.** The experimental setup feels somewhat dated by using GPT-2 Large as the primary model. While the authors include some results with GPT-Neo 2.7B and Pythia-2.8B in the appendix, evaluating more recent models like Phi-3.5 or Llama 3.2 would better demonstrate the method's relevance to current architectures.
>
> **A3.** Thanks for your constructive suggestion. We use Llama-3.2-3B as the base model to conduct the MetaICL experiments, and the results are presented in the table below.  Due to limited computational resources, we provide results for the HR→LR setting. The results show that InvICL continues to outperforms all the baselines, demonstrating its compatibility with the latest model architectures.
>
> | Method | All target tasks | Target tasks in unseen domains |
> | --- | --- | --- |
> | AR ICL | 52.2 | 37.5 |
> | Prefix ICL | 51.6 | 32.1 |
> | PCW (BoE ICL) | 50.9 | 33.4 |
> | InvICL | **53.2** | **39.2** |
>
> We will include the relevant results in the appendix after completing all the experiments.
>
> ---
>
> **Q4.** There are some inconsistencies in the presentation. Figure 1 shows results for PCW while Table 1 uses different terminology (Inv ICL), making it difficult to track the method comparisons. It confuses me to understand the relative performance of different approaches.
>
> **A4.** In fact, these are two different approaches. PCW is a method proposed by previous work, while InvICL is our method. We have added the citation to PCW in the caption of figure 1 to improve clarity.
>
> ---
>
> **Q5.** How would InvICL perform in few-shot settings without MetaICL finetuning? This would be particularly relevant for scenarios where finetuning isn't practical.
>
> **A5.** Please see A1.
>
> ---
>
> Thanks again for your careful reading and detailed review. Please let us know if there is more to clarify. We are happy to take your further question in the discussion stage.

---

> > ### Comment · Reviewer_p8nF · 2024-11-27
> > **Thank you**
> >
> > I appreciate the authors' reply, and they address most of my concerns. I tend to keep my current positive score.

---

### Official Review · Reviewer_k8s4 · 2024-11-03

**Soundness:** 2
**Presentation:** 3
**Contribution:** 3
**Rating:** 6
**Confidence:** 4

**Summary:**

This paper proposes a new permutation-invariant attention mechanism InvICL, which is adapted from Bag-of-Example (BoE) but admits full context-interdependnce. The authors enable the context interdependence by duplicating the context examples and allowing leave-one-out (LOO) attention from the duplicated tokens to original tokens. By doing so, IncICL enjoys the properties of (i) permutation-invariant, (ii) information non-leakage, (iii) context interdependence. Evaluations show that InvICL outperform baselines (Prefix, AR, BoE, NoPE) on tasks that adimit a permutation-invariant nature.

**Strengths:**

The idea is novel and interesting. The motivation is clear as InvICL is proposed by the desired three properties.

**Weaknesses:**

1. Experiment results need more analysis and interpretations. The authors find InvICL shows better length-generalization capabilities, whose mechanism is unclear to me.

2. More experiment results needed. Most results do not have a  reported std. Besides, it would be benificial if there is a figure of squared error curves where the x-axis is the training epochs. Currently there are only results from 50k and 200k epochs.

3. The results of Prefix ICL for linear regression are a bit weird to me. The reported squared error seems to fairly high (around 0.5) even when the number of examples is 40. In another paper [1], the reported squared errors are much lower than 0.1 for 5-layer transformers with number of context examples 10 and data dimension 10.

[1] Johannes von Oswald, Eyvind Niklasson, Ettore Randazzo, João Sacramento, Alexander Mordvintsev, Andrey Zhmoginov, Max Vladymyrov. Transformers Learn In-Context by Gradient Descent. ICML 2023.

**Questions:**

1. Is the loss taken over only the test token or all tokens (autoregressive loss) in the input for Prefix ICL in the training stage? If it is the latter, then it does not make too much sense since for Prefix ICL the context example can attend to its label directly.

2. Can the test token attend to itself in authors' implementation in Prefix ICL for linear regressions? From figure 2 it seems to be attending to itself, but I am not sure if that is the case in the experiments. Other papers [1, 2] using Prefix attention avoid such attention since it could provide incorrect signals to multilayer transformers (the corresponding $\hat y$ is inaccurate within the forward passes).

3. Does the positive encoding in the footnote of page 1 refer to positional encoding?


[1] Johannes von Oswald, Eyvind Niklasson, Ettore Randazzo, João Sacramento, Alexander Mordvintsev, Andrey Zhmoginov, Max Vladymyrov. Transformers Learn In-Context by Gradient Descent. ICML 2023.

[2] Kwangjun Ahn, Xiang Cheng, Hadi Daneshmand, Suvrit Sra. Transformers learn to implement preconditioned gradient descent for in-context learning. NeurIPS 2023.

---

> ### Author Response · Authors · 2024-11-24
> **Response to reviewer k8s4 (Part 1/2)**
>
> We thank reviewer k8s4 for appreciating the novelty of our work. We now address your concerns below.
>
> ---
>
> **Q1.** Experiment results need more analysis and interpretations. The authors find InvICL shows better length-generalization capabilities, whose mechanism is unclear to me.
>
> **A1.** We reckon that the mechanism primarily stems from InvICL achieving invariance. As mentioned in the introduction (Line 38), previous studies have found that **respecting data symmetry in models helps improve generalization**. For example, [1] demonstrated that when the input data exhibits invariance under certain transformations (such as rotation or translation), utilizing an invariant classifier can achieve lower generalization error compared to a regular classifier. [2, 3] concluded that encoding invariances into model improves the effective number of samples, thereby enhance generalization ability. These theoretical results could help explain why InvICL demonstrates stronger length generalization ability.
>
> We have included the above discussion in section 6.
>
> [1] Sokolic, Jure, et al. "Generalization Error of Invariant Classifiers." *arxiv preprint arxiv:1610.04574* (2016).
>
> [2] Bietti, Alberto, Luca Venturi, and Joan Bruna. "On the Sample Complexity of Learning under Invariance and Geometric Stability." *arxiv preprint arxiv:2106.07148* (2021).
>
> [3] Tahmasebi, Behrooz, and Stefanie Jegelka. "The Exact Sample Complexity Gain from Invariances for Kernel Regression." *arxiv preprint arxiv:2303.14269* (2023).
>
> ---
>
> **Q2.** More experiment results needed. Most results do not have a reported std. Besides, it would be beneficial if there is a figure of squared error curves where the x-axis is the training epochs. Currently there are only results from 50k and 200k epochs.
>
> **A2.** Following your suggestion, we
>
> 1. reported std in both synthetic and real-world experiments in the main text (see Figure 3 and Table 2). The results show that the std of the experiments is generally low. In the “Target tasks in unseen domain” setting of the MetaICL experiment, the mean of InvICL is 48.4 with a standard deviation of 1.72, while the mean of AR ICL is 43.6 with a standard deviation of 1.65. Under the normality assumption, hypothesis testing indicates that with 95% confidence, we can conclude that InvICL performs better than AR ICL.
> 2. added Figure 8 and 9 to illustrate how the squared error changes with training epochs. For ease of reading, we present the model’s error when the number of context examples is 100 in the table below (corresponding to Figure 9). The experiments demonstrate that InvICL’s OOD in-context performance (length > 40) consistently outperforms AR ICL across all epochs. Specifically, in the early stages of training, the error of InvICL decreases rapidly, while the error of AR ICL only shows significant reduction after approximately 100k epochs. Furthermore, after 200k epochs, the error of InvICL stabilizes, whereas the error of AR ICL increases.
>
> | Squared error \ Epochs | 20K | 50K | 80K | 100K | 120K | 150K | 180K | 200K | 300K | 400K | 500K |
> | --- | --- | --- | --- | --- | --- | --- | --- | --- | --- | --- | --- |
> | AR ICL | 0.124 | 0.081 | 0.046 | 0.028 | 0.020 | 0.018 | 0.026 | 0.012 | 0.012 | 0.017 | 0.012 |
> | InvICL | 0.991 | 0.989 | 0.915 | 0.809 | 0.359 | 0.112 | 0.111 | 0.141 | 0.182 | 0.236 | 0.270 |
>
> ---

---

> ### Author Response · Authors · 2024-11-24
> **Response to reviewer k8s4**
>
> **Q3.** The results of Prefix ICL for linear regression are a bit weird to me. The reported squared error seems to fairly high (around 0.5) even when the number of examples is 40. In another paper [1], the reported squared errors are much lower than 0.1 for 5-layer transformers with number of context examples 10 and data dimension 10.
>
> **A3.** This is because our experiment follows the setup of [2], which is different from the paper you referred [1]. We choose this setup for the following reasons.
>
> 1. The ICL framework of [2] is more commonly adopted by recent researches (e.g., [3-5]).
> 2. [2] uses a standard (softmax) Transformer, which is more realistic compared to the Transformer with only linear self-attention layers used in [1].
> 3. [2] treats samples and labels as separate tokens, with the input formatted as (x_1, y_1, …, x_n, y_n). This is more realistic compared to [1], where samples and labels are merged and input as ((x_1, y_1), …, (x_n, y_n)).
>
> It is due to the reasons mentioned above that we adopted the setting from [2] in our experiments, leading to results different from those in [1].
>
> [1] von Oswald, Johannes, et al. "Transformers learn in-context by gradient descent." *arXiv preprint arXiv:2212.07677* (2022).
>
> [2] Garg, Shivam, et al. "What Can Transformers Learn In-Context? A Case Study of Simple Function Classes." *arXiv preprint arXiv:2208.01066* (2022).
>
> [3] Li, Yingcong, et al. "Transformers as Algorithms: Generalization and Stability in In-context Learning." *arXiv preprint arXiv:2301.07067* (2023).
>
> [4] Zhang, Ruiqi, Spencer Frei, and Peter L. Bartlett. "Trained transformers learn linear models in-context." *arXiv preprint arXiv:2306.09927* (2023).
>
> [5] Tong, William L., and Cengiz Pehlevan. "MLPs Learn In-Context." *arXiv preprint arXiv:2405.15618* (2024).
>
> ---
>
> **Q4.** Is the loss taken over only the test token or all tokens (autoregressive loss) in the input for Prefix ICL in the training stage? If it is the latter, then it does not make too much sense since for Prefix ICL the context example can attend to its label directly.
>
> **A4.** As explained in section 5.2 (line 402), the loss is computed only on the token to be predicted, i.e., $y_{k+1}$. Therefore, the context example cannot attend to its label directly, thus avoiding the issue of information leakage.
>
> ---
>
> **Q5.** Can the test token attend to itself in authors' implementation in Prefix ICL for linear regressions? From figure 2 it seems to be attending to itself, but I am not sure if that is the case in the experiments. Other papers [1, 2] using Prefix attention avoid such attention since it could provide incorrect signals to multilayer transformers (the corresponding y^ is inaccurate within the forward passes).
>
> **A5.** Yes, in our implementation of Prefix ICL, the test tokens can attend to themselves.
>
> In fact, the modeling in [1,2] differs somewhat from real-world scenarios. Specifically, they consider the input in the form of $((x_1, y_1), \dots, (x_n, y_n), (x_t, 0))$, where $(x_i, y_i), i \in [n]$ represents the samples and labels of the context examples, and $x_t$ represents the test sample. Their goal is to encode $(x_t, 0)$ into $(x_t, y_t)$ through Transformer to obtain the prediction, rather than generating $y_t$ as the next token. To prevent the test token from accessing its own label, they block self-attention for these tokens.
>
> In contrast, under general circumstances, the sample x and label y in ICL typically correspond to different tokens. In other words, the input follows the form of $(x_1, y_1, \dots, x_n, y_n, x_t)$. Therefore, the label information is not leaked during the generation of $y_t$. Consequently, there is no need to prevent self-attention for the test tokens. This is why we use this form of Prefix ICL, rather than the form described in [1,2].
>
> **Q6.** Does the positive encoding in the footnote of page 1 refer to positional encoding?
>
> **A6.** Yes, it does. We have fixed it in the manuscript.
>
> ---
>
> Thanks again for your constructive suggestions. Please let us know if there is more to clarify.

---

> ### Comment · Reviewer_k8s4 · 2024-11-24
> **Is Section 4 necessary?**
>
> I see. Thanks for your response. I previously had some misunderstanding on the input format because of Section 4, where you invoke the setting and the construction from [1]. Now I feel this section a bit overclaiming since there's no experimental evidence justifying that InvICL is approximately performing GD as it is merely a construction. In [1] they have the claim because they indeed found the strong evidence in the heatmap of the weights matrices. Since you are taking a totally different input and model setting in the experiments, I suspect it would be hard to observe any weights pattern (please correct me if I am wrong). Actually I personally feel it is not necessary to justify section 4, instead, it might be better to make the claim in section 4 weaker and move it to the appendix. This should not affect your main contribution on the architecture design and the three desired properties. I would like to increase my score if section 4 is better placed.
>
> [1] von Oswald, Johannes, et al. "Transformers learn in-context by gradient descent." arXiv preprint arXiv:2212.07677 (2022).

---

> > ### Author Response · Authors · 2024-11-25
> >
> > Thanks for your constructive suggestion. We have moved section 4 (theoretical results) into the appendix and adjust the expression in the newest version.

---

> > > ### Comment · Reviewer_k8s4 · 2024-11-25
> > > **Have the changes been made?**
> > >
> > > Sorry I may be missing something, but I didn't notice anything significantly different in the newest pdf. Has it been updated?

---

> > > > ### Author Response · Authors · 2024-11-25
> > > >
> > > > Sorry, we didn’t submit the updated version correctly earlier. The new version has now been updated.

---

> > > > > ### Comment · Reviewer_k8s4 · 2024-11-25
> > > > >
> > > > > Thanks. I have increased my rating to 6.

---

> > > > > > ### Author Response · Authors · 2024-11-25
> > > > > >
> > > > > > Thanks for improving the score! We are delighted that our elaborations and revisions have addressed your concerns. Have a great day!

---

### Author Response · Authors · 2024-11-24
**Updates in the manuscript**

Following the suggestion from the reviewers, we have updated a revised manuscript with the main changes highlighted in orange, which are:

- **Add new experiments results.**
    1. In Appendix B.4, we added figures to illustrate how the squared error changes with training epochs under the synthetic setting.
    2. In Appendix B.5, we added linear probing experiments to explore how the architecture of InvICL impacts the model’s internal representations.
    3. We added std for figure 3 and table 2.

- **Add more discussion.** In section 6, we added discussion on
    1. The mechanism behind InvICL’s strong length generalization ability.
    2. Theoretical complexity of InvICL.
    3. Practicality of Theorem 4.1.
    4. The ICL training objective.

- **Improve presentation.**
    1. We added references to the methods used in Figure 1 for clarity.

---

> ### Author Response · Authors · 2024-11-25
>
> Following the suggestion of Reviewer k8s4, we have moved section 4 (theoretical results) into the appendix and adjust some of the expression to describe the contributions more accurately. The discussion about the practicality of theorem 4.1 is moved to the appendix accordingly.

---

### Meta-Review · Area_Chair_XvDw · 2024-12-19

**Metareview:**

This paper proposes three properties that ideal ICL algorithms should satisfy (invariance, information non-leakage and context interdependence), and developed a new InvICL method to achieve the three simultaneously. Reviewers agree that the paper is well written and the empirical results are interesting. Although there were some clarifying questions and concerns about computation for the new method, most of the issues are addressed during author response.

**Additional Comments On Reviewer Discussion:**

The reviewers have a uniform recommendation and agree with each other.

---

### Decision · Program_Chairs · 2025-01-22

Accept (Poster)